# DC-Spin: A Speaker-invariant Speech Tokenizer for Spoken Language Models

## Abstract

Spoken language models (SLMs) have gained increasing attention with advancements in text-based, decoder-only language models. SLMs process text and speech, enabling simultaneous speech understanding and generation. This paper presents Double-Codebook Speaker-invariant Clustering (DC-Spin), which aims to improve speech tokenization by bridging audio signals and SLM tokens. DC-Spin extracts speaker-invariant tokens rich in phonetic information and resilient to input variations, enhancing zero-shot SLM tasks and speech resynthesis. We propose a chunk-wise approach to enable streamable DC-Spin without retraining and degradation. Comparisons of tokenization methods (self-supervised and neural audio codecs), model scalability, and downstream task proxies show that tokens easily modeled by an n-gram LM or aligned with phonemes offer strong performance, providing insights for designing speech tokenizers for SLMs.

## 1 Introduction

Spoken language models (SLMs) and related applications have gained more interest with the advancements of large language models (LLM) and audio tokenization techniques (Wu et al., 2024). These speech LMs resemble causal LMs in natural language processing, but SLMs take speech and, optionally, text as input and generate speech or text. Hence, these LMs can perform tasks like speech continuation (Lakhotia et al., 2021), automatic speech recognition (ASR) (Rubenstein et al., 2023; Maiti et al., 2024), text-to-speech synthesis (TTS) (Wang et al., 2023), and the more complicated spoken language understanding (SLU) problems (Gong et al., 2023; Chu et al., 2023; Nguyen et al., 2024). SLM has two main research directions: 1) LM architecture and training and 2) speech tokenization techniques, the latter of which is the focus of this paper.

Since directly taking raw audio waveform as input to an SLM is infeasible, tokenizing speech into text-like discrete units has become an essential component of recent SLMs. We define four key qualifications for a good speech tokenizer inspired by prior studies. First, the tokens should contain strong phonetic or semantic information so that the SLM can use the content of speech to perform ASR and SLU (Lakhotia et al., 2021). Second, the tokens should retain acoustic details for being resynthesized into speech for generative tasks like TTS and speech-to-speech translation (Lee et al., 2022; Zhang et al., 2024; Wang et al., 2023). Third, the tokenizer should be robust to perturbations like additive noise, reverberation, and speaker change because the perturbations are irrelevant to how an SLM understands human speech and language (Gat et al., 2023; Messica & Adi, 2024). Fourth, the tokenizer should be lightweight and fast, supporting real-time interaction between users and SLMs. Hence, this paper tries to answer the following question: *how to build and evaluate a good speech tokenizer for spoken language models that satisfies these key qualifications?*

We simplify the setup of this paper by training a unit-based speech LM (uLM) (Lakhotia et al., 2021) and a Hifi-GAN unit-to-speech synthesizer (Kong et al., 2020; Polyak et al., 2021). This setup is commonly used in SLM studies and applications (Maiti et al., 2024; Messica & Adi, 2024; Hassid et al., 2024), which is an ideal proxy for more advanced SLMs. uLMs are decoder-only transformer LMs (Vaswani et al., 2017) and trained with the next-token prediction objective on speech tokens. uLMs can perform zero-shot tasks by estimating the probability of utterances, including detecting real spoken words and determining correct syntactic structures (Nguyen et al., 2020), and can be fine-tuned for ASR. Moreover, we train Hifi-GANs to convert tokens to audio and quantify the intelligibility of the resynthesized speech to simulate speech generation with SLMs. With uLM and

resynthesis, we can examine speech tokenizers on the first two required qualities. Next, we follow Gat et al. (2023) to quantify the robustness by comparing the extracted tokens between clean and perturbed speech. Finally, we measure the inference speed of offline and streaming tokenization.

After defining the goals and evaluation pipelines, we propose Double-Codebook Spin (**DC-Spin**) by extending speaker-invariant clustering (Spin) with an auxiliary codebook to extract better speech units, where Spin is a self-supervised fine-tuning method for capturing phonetic units via online clustering and speaker-invariant swapped prediction (Chang et al., 2023). To further boost robustness and token quality, we propose pre-training the Hidden-unit BERT (HuBERT) self-supervised speech encoder with Spin codeword units as a better initialization for DC-Spin (Hsu et al., 2021), denoted as **SpinHuBERT**. The contributions of this paper are listed as follows:

1. The proposed speech tokenizer produces high-quality speaker-invariant speech tokens, achieving state-of-the-art spoken language modeling and speech resynthesis compared to open-source tokenizers on multiple benchmarks with limited resources.
2. We propose a simple chunk-wise method to repurpose offline speech tokenizers into streaming mode with a negligible performance drop.
3. We analyze multiple proxy tasks to understand the relation between speech tokenizer and SLM performance. We find that phoneme and character-normalized mutual information and the proposed n-gram predictability are good proxies for downstream tasks.

## 2 RELATED WORK

**Spoken Language Models (SLM)**  SLMs or speech language models usually refer to decoder-only LMs that input or output speech and sometimes text. The two main approaches to integrating speech into LMs are adaptor and token-based.[1] Because of the recent advancements in LLMs, researchers connect speech encoders and text-based LMs through adaptors (Chu et al., 2023; Ma et al., 2024; Tang et al., 2024), allowing speech understanding and ASR but requiring a more sophisticated design for speech generation (Dubey et al., 2024). In contrast, a more common approach, which is the main focus of this paper, is to tokenize speech to serve as both input and output of SLMs (Lakhotia et al., 2021; Hassid et al., 2024; Maiti et al., 2024). Under this setup, SLMs treat audio waveforms as text-like tokens, allowing SLMs to process speech and text jointly (Nguyen et al., 2024) and to generate speech by synthesizing tokens into audio (Polyak et al., 2021).

In Lakhotia et al. (2021), SLMs are trained with unlabeled speech tokens to discover the spoken content, which can be evaluated with zero-shot tasks in Nguyen et al. (2020). This concept allows an SLM to be fine-tuned with paired speech-text data for ASR, TTS, and SLU (Maiti et al., 2024). Advanced techniques like interleaving speech and text tokens (Nguyen et al., 2024), initializing with text-based LMs (Hassid et al., 2024), and integrating multiple token types (Borsos et al., 2023) are developed to improve performance. Because SLMs simultaneously understand and generate speech, and the speech tokens are the only media between the models and audio signals, speech tokenizer design has become a crucial part of SLM research.

**Self-supervised Learning (SSL)**  SSL is introduced to leverage large unlabeled audio datasets to pre-train speech encoders, mitigating the need for extensive human labeling (Mohamed et al., 2022). SSL models are trained to predict pseudo labels given a partial speech utterance. Pseudo targets could be Mel spectrograms (Chung et al., 2019; Liu et al., 2021), vector-quantized features (Baevski et al., 2020; Hsu et al., 2021; Chiu et al., 2022), or an exponential average of the model itself (Baevski et al., 2022; Liu et al., 2023). Pre-trained SSL models offer good initialization for speech processing tasks (Yang et al., 2024). Moreover, evidence has shown that speech SSL models excel at extracting phonetic representations (Pasad et al., 2021; Chang et al., 2023; Choi et al., 2024), so quantizing SSL hidden layer embeddings with K-means clustering is widely adopted to tokenize speech (Lakhotia et al., 2021; Hassid et al., 2024; Maiti et al., 2024). Gat et al. (2023) and Messica & Adi (2024) further fine-tune SSL encoders for robust speech tokenizers.

---

[1]Terms "token" and "unit" are used interchangeably in this paper, indicating discrete speech units.

**Neural Audio Codec**  Neural network-based codecs compress audio into compact units and reconstruct high-fidelity signals from the units (Zeghidour et al., 2021; Défossez et al., 2023; Wu et al., 2023). These models resemble autoencoders and comprise an encoder, a quantization module, and a decoder. A commonly used technique for the quantization module is residual vector quantization (RVQ) (Zeghidour et al., 2021). RVQ has multiple codebooks, each quantizing the residual features computed from the previous codebook, making the first few codebooks preserve more critical information for reconstructing audio waveforms. Zhang et al. (2024) proposes SpeechTokenizer by enforcing the first codebook to capture phonetic units by distilling knowledge from a pre-trained SSL teacher, but the teacher bounds the performance. One of the benefits of neural codecs is that the model itself has an audio resynthesis module, i.e., the decoder. Still, SSL-based tokenizers can resynthesize speech with a separate vocoder.

Besides the open-source tokenizers, closed models like USM (Rubenstein et al., 2023) are claimed to be powerful for SLMs, but these tokenizers are difficult to reproduce or compare because the details remain unrevealed. In contrast, this paper aims to offer insights into designing tokenizers and shares all details for future studies. Additionally, some works categorize speech tokens into semantic and acoustic tokens for understanding and generative tasks, respectively (Zhang et al., 2024; Borsos et al., 2023). However, we will demonstrate that a single type of speech token is sufficient to perform well on both tasks.

## 3 METHOD

### 3.1 BACKGROUND

**Speaker-invariant Clustering (Spin)**  Spin is a self-supervised fine-tuning approach inspired by Caron et al. (2020) and captures speaker-invariant content in speech signals through online clustering and swapped prediction (Chang et al., 2023). During training, each utterance is perturbed to sound like a different speaker but with the same content by randomly scaling the F0 and formant frequencies. Both utterances are fed to a pre-trained SSL encoder, and the frame-level output of each utterance is transformed into a sequence of probability distributions with a learnable codebook. The distributions are smoothed to enforce full codebook usage and serve as the learning target. Finally, the model performs swapped prediction by minimizing the cross-entropy loss between the original codeword distribution and the smoothed targets from the perturbed output and vice versa.

Spin efficiently improves SSL encoders in content-related problems like ASR and phoneme recognition (PR). Robust-Spin (R-Spin) extends Spin for robust speech recognition but requires more complicated training stages and implementation (Chang & Glass, 2024). Although Chang & Glass (2024) have shown that discrete units produced by Spin codebooks are closely aligned with phonemes and characters, the applications of these tokens remain undiscovered.

**Hidden-unit BERT (HuBERT)**  HuBERT is an SSL pre-training method for speech representation learning (Hsu et al., 2021). Like BERT in NLP (Devlin et al., 2019), HuBERT is pre-trained with a mask prediction objective for multiple iterations with pseudo labels derived by K-means clustered continuous audio representations. First, the labels are K-means cluster IDs of Mel-frequency cepstral coefficients (MFCCs). Then, the second iteration model predicts K-means clusters from the first model's hidden embeddings. Besides serving as pre-training labels, K-means units are useful in SLM and speech-to-speech translation (Lakhotia et al., 2021; Lee et al., 2022). This paper adopts HuBERT as the initialization of the proposed speech tokenizers (Section 3.2) and further improves HuBERT by introducing better learning targets (Section 3.3).

### 3.2 SPIN AS SPEECH TOKENIZER

This section proposes tokenizing speech with Spin codebook along with methods to improve the quality of Spin discrete units. Because Spin codebooks capture phonetic information and have a unique speaker-invariant property, the tokens extracted from Spin satisfy the first qualification in Section 1. These properties are especially useful for speech generation because the vocoder can condition on different speakers, allowing more flexible speech synthesis. Compared with K-means, Spin's codebook is optimized with gradient descent, proven highly scalable (LeCun et al., 2002). In contrast, K-means clustering requires extracting and storing hidden features, leading to high memory consumption and special implementation when scaling (Zanon Boito et al., 2024).

Figure 2: The proposed multi-stage training for the DC-Spin (Section 3.2). Stage (I) pre-trains a speech encoder with pseudo labels from K-means or Spin units, where the latter is the proposed SpinHuBERT (Section 3.3). The optional stage (II) fine-tunes the encoder with CTC-based ASR or phoneme recognition (PR). In stage (III), the encoder is fine-tuned with DC-Spin to obtain the codebook for extracting discrete speech tokens.

Furthermore, K-means tokens contain speaker and unrelated information, leading to suboptimal SLM performance (Yeh & Tang, 2024). Motivated by the above reasons, this paper explores the possibilities of tokenizing speech with Spin for SLMs.

First, we fine-tune HuBERT Base with different Spin codebook sizes and use the codeword IDs as discrete units to perform zero-shot spoken LM tasks, where the experimental setup can be found in Section 4.1. As shown in Figure 1, ideal codebook sizes are between 200 and 500. Note that the codebook size should be large enough for speech resynthesis since low bitrate degrades resynthesis quality (Appendix D). Moreover, Chang et al. (2023) found larger Spin codebooks capture better phonetic representations in the encoder. The contradictory properties motivate us to develop methods to obtain a small but high-quality codebook.

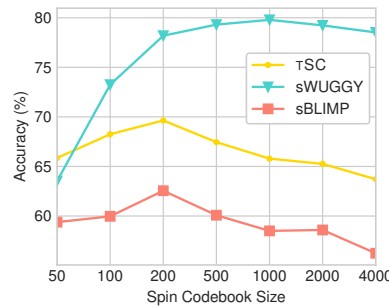

Figure 1: HuBERT + Spin tokenizers on zero-shot SLM (see Section 4.2).

**Double-Codebook Spin (DC-Spin)**   DC-Spin extends Spin to two learnable codebooks optimized with the same objective. The first codebook (primary) extracts discrete units for downstream applications. The second codebook (auxiliary) is a large codebook that enhances the encoder's capability to capture fine-grained phonetic units. Because both codebooks share the same encoder, the auxiliary codebook is expected to indirectly help the primary codebook encode high-quality units.

**Supervised Fine-tuning (SFT)**   Inspired by the speech encoders in multimodal LLMs (Rubenstein et al., 2023; Gemini Team, 2023; Dubey et al., 2024), we include supervised fine-tuning to boost the token quality. Specifically, we consider CTC-based (Graves et al., 2006) ASR and PR as the supervised tasks because 1) the data for these objectives are relatively easy to collect compared to frame-wise labels and 2) both tasks force the model to neglect redundant information and extract the content in speech. CTC fine-tuning can be applied before or during DC-Spin fine-tuning, but we found the former leads to better results (Appendix B.3).

### 3.3 HuBERT Pre-training with Better Targets

Spin can be applied to any pre-trained speech encoder, but the fine-tuned performance depends on the encoder's quality. In Table 1, HuBERT and data2vec are superior to other methods, even though all models are fine-tuned with the same DC-Spin objective. HuBERT is slightly inferior to data2vec on two tasks, but data2vec is more unstable because the learning target is an exponential moving average of it-

Table 1: Zero-shot SLM accuracy for DC-Spin with SSL encoders sharing similar architectures. See Section 4.2 for task descriptions.

| SSL Model | TSC | sWUGGY | sBLIMP |
|---|---|---|---|
| wav2vec 2.0 (Baevski et al., 2020) | 66.0 | 75.7 | 55.4 |
| HuBERT (Hsu et al., 2021) | 67.5 | **81.4** | 60.8 |
| data2vec (Baevski et al., 2022) | **68.8** | 67.6 | **64.9** |
| DinoSR (Liu et al., 2023) | 65.7 | 62.1 | 61.3 |

self (Baevski et al., 2022). In contrast, HuBERT has fixed learning targets, which can be replaced with better pseudo labels. The above findings have led us to propose **SpinHuBERT** by training HuBERT models with labels Spin units to better initialize DC-Spin. Because of the speaker-invariant nature of Spin, Chang et al. (2023) and (Chang & Glass, 2024) have shown that discrete units derived from Spin codebooks are closer to phonetic units than HuBERT K-means units. Following this observation, SpinHuBERT is expected to extract better phonetic representations.

Summarizing the proposed DC-Spin and SpinHuBERT, the training pipeline is shown in Figure 2. In stage (I), we pre-train a SpinHuBERT encoder with pseudo labels generated with Spin. The optional stage (II) fine-tunes the encoder with CTC-based ASR or PR. Stage (III) fine-tunes the encoder with the proposed DC-Spin objective to obtain the discrete speech tokens for downstream applications.

## 4 EXPERIMENT

### 4.1 SETUP

**Baseline Tokenizers**  We adopt EnCodec 24kHz (Défossez et al., 2023) and SpeechTokenizer (Zhang et al., 2024) as the neural audio codec baselines.[2]  For SSL-based methods, we consider K-means clustering, augmentation invariant discrete representation (Gat et al., 2023), and Noise Aware Speech Tokenization (NAST) (Messica & Adi, 2024), where the second and third methods are designed specifically for SLM by training with perturbation-invariant objectives.[3] An SSL-based tokenizer using K-means clustering with $K$ units is denoted as "K-means$_K$."

**Self-supervised Pre-training**  The HuBERT models are trained for 400k steps with 124k hours of unlabeled English speech. Following the Large and X-Large models in Hsu et al. (2021), our 3rd-iteration HuBERT (it3) learns to predict 500-unit K-means clusters of the 9th layer of HuBERT Base. SpinHuBERT learns from a Spin model with a codebook size of 4096. Unless specified otherwise, SSL models operate at a 50Hz framerate. Details can be found in Appendix A.2.

**Supervised Fine-tuning**  Under the SFT setup, we fine-tune pre-trained SSL models with ASR and PR before applying DC-Spin using two labeled datasets: LibriSpeech and English Labeled 3k. The latter extends LibriSpeech with an additional 2k hours of speech. The fine-tuned encoders are denoted by appending "ASR$_{nk}$" and "PR$_{nk}$" to the encoder's name, where $n = 1$ or $3$, indicating the two dataset sizes. See Appendix A.3 for more information.

**Spin & DC-Spin Fine-tuning**  We follow Chang et al. (2023) and reimplement Spin in fairseq (Ott et al., 2019). We fine-tune SSL models with unlabeled data from LibriSpeech on a single NVIDIA 32GB V100 GPU (see Appendix A.4). "Spin$_K$" denotes Spin with a codebook size of $K$. DC-Spin with primary and auxiliary codebook sizes of $K_1$ and $K_2$ is denoted as "DC-Spin$_{K_1,K_2}$."

**Spoken Language Models**  We adopt unit-based LM as the SLM for a fair comparison with prior works (Lakhotia et al., 2021). Each SLM is a 150M-parameter transformer decoder (Vaswani et al., 2017) that performs next-token prediction on discrete speech units. The training data are obtained by extracting units from the 6k hours clean subset of Libri-Light (Kahn et al., 2020). After training, SLM estimates the log probability of speech utterances normalized by length for zero-shot SLM tasks. Furthermore, we fine-tune SLMs with the same training objective but with labeled data from LibriSpeech to perform ASR. See Appendix A.5 for more details.

**Speech Resynthesis**  We use the Expresso dataset (Nguyen et al., 2023) to train and evaluate unit-to-speech Hifi-GAN vocoders (Kong et al., 2020; Polyak et al., 2021). The input includes a sequence of tokenized speech units, a speaker ID, and a style ID. After training, we resynthesize all utterances in the dev and test sets with the original speaker and style IDs.

In our experiments, the speech tokens are deduplicated for SLM, i.e., merging repeated consecutive tokens, so the SLM outputs are also deduplicated, requiring the vocoder to include a duration prediction module in real-world applications.[4] However, to avoid further uncertainties, we simplify the vocoder setup to take speech token sequences with the correct length as input (Chang et al., 2024). We keep the SLM and vocoder simple to reduce the effects of downstream model design and amplify

---

[2]`https://github.com/ZhangXInFD/SpeechTokenizer`

[3]`https://github.com/ShovalMessica/NAST`

[4]E.g., a token sequence `45 103 103 34 5 5 5` after deduplication would be `45 103 34 5`.

Table 2: Zero-shot SLM evaluation for unsupervised speech tokenizers based on HuBERT Base and the LibriSpeech dataset. All SLMs share the same architecture (150M parameters).

| Units | Method | TSC↑ | sWUGGY↑ | | sBLIMP↑ |
|---|---|---|---|---|---|
| | | | all | in-vocab | |
| 50 | K-means♠ | 66.27 | – | 67.48 | 52.42 |
| | Gat et al. (2023)♣ | – | – | 67.42 | 57.04 |
| | NAST$_{50}$ (Messica & Adi, 2024) | 64.51 | – | 67.14 | 54.34 |
| | NAST$_{50}$ (Messica & Adi, 2024)◇ | 67.13 | 61.62 | 67.35 | 55.68 |
| | Spin$_{50}$ | 65.85 | 58.90 | 63.52 | 59.38 |
| | DC-Spin$_{50,4096}$ | **69.91** | **65.05** | **73.51** | **60.15** |
| 100 | K-means♠ | 67.18 | – | 67.75 | 51.96 |
| | Gat et al. (2023)♣ | – | – | 68.20 | 56.99 |
| | NAST$_{100}$ (Messica & Adi, 2024) | 64.13 | – | 73.35 | 55.86 |
| | NAST$_{100}$ (Messica & Adi, 2024)◇ | 66.70 | 65.14 | 71.99 | 56.09 |
| | Spin$_{100}$ | 68.25 | 65.28 | 73.25 | 59.97 |
| | DC-Spin$_{100,4096}$ | **70.18** | **68.04** | **78.47** | **61.35** |
| 200 | K-means♠ | 67.55 | – | 71.88 | 52.43 |
| | Gat et al. (2023)♣ | – | – | 70.68 | 56.26 |
| | NAST$_{200}$ (Messica & Adi, 2024) | 66.70 | – | 76.42 | 55.62 |
| | NAST$_{200}$ (Messica & Adi, 2024)◇ | 67.88 | 63.63 | 70.45 | 53.45 |
| | Spin$_{200}$ | **69.64** | 68.95 | 78.19 | **62.55** |
| | DC-Spin$_{200,4096}$ | 69.21 | **70.79** | **80.59** | 62.13 |
| 500 | K-means | 63.23 | 66.74 | 74.72 | 55.54 |
| | Gat et al. (2023)♣ | – | – | 69.33 | 56.93 |
| | Spin$_{500}$ | 67.45 | 70.03 | 79.31 | 60.08 |
| | DC-Spin$_{500,4096}$ | **67.50** | **71.48** | **81.38** | **60.84** |

♠Source: Messica & Adi (2024). ◇Reproduced with official checkpoints.
♣The authors could not confirm the subset of sWUGGY they reported, but it is more likely to be the in-vocab set according to Messica & Adi (2024).

the impact of tokenizers since this paper aims to understand how to design speech tokenizers and how they affect SLM performance. The applications can be extended by introducing more advanced modeling strategies, but we leave this part for future studies.

## 4.2 ZERO-SHOT SPOKEN LANGUAGE MODELING

This section discusses the impact of tokenizers on SLM by adopting the following tasks.
**TSC** We use the "Topic" Spoken StoryCloze to evaluate an SLM's ability to capture continuation coherence and fine-grained textual nuances (Hassid et al., 2024). Each sample comprises two similar spoken stories with different endings. The SLM must find the utterance with a consistent ending.
**sWUGGY** We adopt the sWUGGY spot-the-word task from ZeroSpeech (Nguyen et al., 2020).[5] Each sample has two spoken words with similar pronunciations, with one of the words absent from the English vocabulary. The "all" subset combines the "in-vocab" subset and out-of-vocabulary words that do not appear in the LibriSpeech training set.
**sBLIMP** The sBLIMP acceptability metric is also adopted from ZeroSpeech. Each sample comprises two similar utterances, but one is ungrammatical. The above tasks require an SLM to compute a pseudo probability for each audio recording in a sample and compare the probabilities to determine which is more likely to be the correct answer. The results are reported in accuracy.

Table 2 shows the results of unsupervised speech tokenization techniques based on HuBERT Base and LibriSpeech for a fair comparison. DC-Spin demonstrates superior performance compared with previous methods. We observe consistent improvement of DC-Spin over Spin across different unit sizes, but the gap is narrowed when the codebook size is 500. Among all tasks, DC-Spin improves sWUGGY most significantly because this problem is closely related to how well speech tokens represent pronunciation, which is directly related to phonetic information. The results strongly indicate the effectiveness of DC-Spin.

---

[5]https://github.com/zerospeech/benchmarks

Table 3: Unconstrained resources zero-shot spoken language modeling results. We use the first RVQ codebook of audio codecs to extract speech tokens.

| Method | SLM Params | SLM Data (hours) | TSC↑ | sWUGGY↑ all | sWUGGY↑ in-vocab | sBLIMP↑ |
|---|---|---|---|---|---|---|
| **High-resource Speech LM** | | | | | | |
| AudioLM (Borsos et al., 2023) | 300M | 60k | – | 71.5 | 83.7 | 64.7 |
| VoxtLM (Maiti et al., 2024)♠ | 1.3B | 60k | – | 65.6 | – | 57.1 |
| TWIST (Hassid et al., 2024)♠ | 1.3B | 150k | 70.6 | 72.7 | 82.5 | 57.0 |
| TWIST (Hassid et al., 2024)♠ | 7B | 150k | 74.1 | 73.9 | 83.6 | 59.0 |
| TWIST (Hassid et al., 2024)♠ | 13B | 150k | 76.4 | **74.5** | 84.1 | 59.2 |
| SpiRit-LM (Nguyen et al., 2024)♠ | 7B | 460k | **82.9** | 69.0 | – | 58.3 |
| **K-means$_{500}$** | | | | | | |
| HuBERT Base | 150M | 6k | 63.2 | 66.7 | 74.7 | 55.5 |
| HuBERT Base@25Hz♣ | 150M | 6k | 66.9 | 68.6 | 78.0 | 56.3 |
| Whisper Small (Radford et al., 2023) | 150M | 6k | 61.2 | 62.5 | 68.5 | 53.9 |
| **Audio Codecs** | | | | | | |
| EnCodec (Défossez et al., 2023) | 150M | 6k | 56.1 | 52.2 | 53.1 | 50.1 |
| SpeechTokenizer (Zhang et al., 2024) | 150M | 6k | 63.7 | 64.9 | 72.1 | 53.9 |
| **SpinHuBERT@50Hz + DC-Spin$_{500,4096}$ (Proposed)** | | | | | | |
| SpinHuBERT | 150M | 6k | 70.7 | 72.3 | 82.2 | 62.8 |
| SpinHuBERT-ASR$_{1k}$ | 150M | 6k | 69.3 | **74.5** | **85.5** | 65.6 |
| SpinHuBERT-PR$_{1k}$ | 150M | 6k | 69.7 | 73.7 | 84.7 | 65.3 |
| SpinHuBERT-ASR$_{3k}$ | 150M | 6k | 70.2 | 73.7 | 84.5 | 65.7 |
| SpinHuBERT-PR$_{3k}$ | 150M | 6k | 70.2 | 74.1 | 85.0 | **65.9** |
| **Cascaded Topline** | | | | | | |
| ASR + Llama2 (Nguyen et al., 2024) | 7B | – | 94.8 | 79.2 | – | 71.6 |

♠LM trained with text or paired speech-text data. ♣The HuBERT model used in Nguyen et al. (2024).

To compare the proposed methods with state-of-the-art SLMs, we report results with unconstrained resources in Table 3. The proposed SpinHuBERT with DC-Spin offers the best performance on sWUGGY and sBLIMP, even using a relatively small SLM and training data size. For TSC, DC-Spin performs similarly with 1.3B-parameter TWIST (Hassid et al., 2024), but the gap increases between DC-Spin and larger SLMs, showing that this task might correlate more with LM scaling, especially when comparing to the cascaded topline. Furthermore, DC-Spin is improved using either ASR or PR SFT with similar performance gains, indicating that either task is suitable for assisting DC-Spin. As for the baselines, the Whisper Small encoder (87M parameters) with K-means offers low accuracy even though the encoder was trained with 680k hours of speech. EnCodec tokens result in the worst performance because no explicit constraints are imposed on the encoder or quantizer to extract phonetic or semantic representations. SpeechTokenizer performs similarly to HuBERT with K-means, corroborating the hypothesis mentioned in Section 2 that the HuBERT teacher bounds this model. Hence, building speech tokenizers from speech SSL models offers better representations for SLM. Overall, the results suggest that speech tokenizers greatly impact SLMs, and the proposed SpinHuBERT and DC-Spin achieve state-of-the-art SLMs on several tasks with limited resources.

## 4.3 SPEECH RESYNTHESIS

This section focuses on speech generation with SLMs by resynthesizing speech from discrete units and evaluating with the following metrics.

**ASR-WER** This metric uses an ASR model to transcribe the resynthesized speech and computes the word error rate (WER) to quantify the intelligibility of the audio.[6]

**UTMOS** Following prior works (Mousavi et al., 2024; Chang et al., 2024), we adopt UTMOS, a neural network-based mean opinion score (MOS) prediction, to assess the quality of the resynthesized speech because this metric highly correlates with human-rated MOS (Saeki et al., 2022). Although other metrics exist to evaluate vocoders, we focus on whether the speech tokens preserve sufficient information to synthesize intelligible and human-like speech using the same vocoder.

---

[6]https://dl.fbaipublicfiles.com/fairseq/wav2vec/wav2vec_vox_960h_pl.pt

As shown in Table 4, HuBERT with DC-Spin reduces more than 10% relative WER compared with K-means, but the K-means and DC-Spin are similar in SpinHuBERT, showing that training Hu-BERT with Spin units helps representations for resynthesis. SFT with 1k hours of data has little impact on the resynthesis results, although SFT has removed some acoustic details. Moreover, SFT with more data (1k vs. 3k hours) lowers ASR-WER, which might be caused by increased robustness. Compared with codec-based approaches, DC-Spin tokens can be synthesized to produce high-intelligibility and quality speech at a relatively low bitrate because the acoustic details are encoded across several RVQ codebooks in codecs. We notice that UTMOS among SSL-based methods are similar, possibly indicating that the resynthesis quality is less relevant to the tokens than the vocoder. To summarize, this section demonstrates the effectiveness of SSL-based tokenizers on speech resynthesis, corroborating with the findings in Shi et al. (2024b).

Table 4: Speech resynthesis ASR-WER and UTMOS on Expresso dev and test sets.

| Method | Bitrate | ASR-WER↓ | | UTMOS↑ | |
|---|---|---|---|---|---|
| | | dev | test | dev | test |
| Ground Truth | 256k | 15.2 | 14.3 | 3.24 | 3.28 |
| **EnCodec (Défossez et al., 2023)** | | | | | |
| RVQ1:2 | 1.5k | 28.4 | 27.5 | 1.35 | 1.31 |
| **SpeechTokenizer (Zhang et al., 2024)** | | | | | |
| RVQ1 | 500 | 30.7 | 32.9 | 1.27 | 1.27 |
| **HuBERT** | | | | | |
| K-means$_{500}$ | 448 | 24.0 | 24.4 | 2.93 | 2.76 |
| DC-Spin$_{500,4096}$ | 448 | 21.3 | 22.4 | 2.96 | 2.93 |
| + ASR$_{1k}$ | 448 | 21.6 | 22.9 | 2.96 | 2.96 |
| + PR$_{1k}$ | 448 | 21.4 | 22.5 | 3.00 | 2.97 |
| **SpinHuBERT** | | | | | |
| K-means$_{500}$ | 448 | 20.0 | 21.2 | 3.05 | 2.94 |
| DC-Spin$_{500,4096}$ | 448 | 20.5 | 21.7 | **3.11** | 3.04 |
| + ASR$_{1k}$ | 448 | 21.7 | 22.6 | 2.90 | 2.84 |
| + PR$_{1k}$ | 448 | 21.0 | 20.7 | 2.93 | 2.84 |
| + ASR$_{3k}$ | 448 | 18.9 | 20.0 | 3.08 | **3.05** |
| + PR$_{3k}$ | 448 | **18.8** | **18.7** | 3.02 | 2.92 |

## 4.4 ROBUSTNESS

This section focuses on the robustness of speech tokenizers via **unit edit distance (UED)** (Gat et al., 2023).[7] This metric computes the unit error rate of speech tokens between clean and distorted audio inputs, so lower values imply superior robustness.

In Table 5, Spin and DC-Spin surpass Gat et al. (2023) under most distortions even though this baseline tokenizer is explicitly trained with a denoising objective while our methods only have a speaker-invariant constraint. One surprising finding is that Spin and DC-Spin are less robust on pitch shift than other distortions,

Table 5: Unit edit distance using 500 units with four types of audio distortions. DC-Spin⋆ is based on SpinHuBERT-PR$_{3k}$.

| Method | Noise | Time Stretch | Reverb | Pitch Shift |
|---|---|---|---|---|
| K-means | 50.6 | 58.9 | 39.7 | 36.5 |
| Gat et al. (2023) | 36.5 | 40.8 | 25.8 | 27.5 |
| Spin | 22.3 | 30.5 | 13.8 | 35.9 |
| DC-Spin | 22.0 | 29.2 | 13.5 | 35.1 |
| DC-Spin⋆ | **13.5** | **21.6** | **11.5** | **24.1** |

probably because the distortion always shifts the pitch by a major third, making the speakers with higher pitches sound unreal. In contrast, the speaker perturbation approach in Spin training keeps the speech more natural (Choi et al., 2021). Moreover, the overall best-proposed SpinHuBERT-PR$_{3k}$ + DC-Spin tokenizer (the last row) reduces the UED values further. Overall, the proposed tokenizers demonstrate robustness even in unseen scenarios.

## 4.5 INFERENCE EFFICIENCY

This section inspects the inference efficiency, the last qualification for good speech tokenizers. The following metrics are averaged over three runs on LibriSpeech dev-clean and dev-other using a single V100 GPU. **Latency** is the average time required to tokenize an utterance. **Real Time Factor (RTF)** is the ratio between latency and utterance duration, so a lower value implies faster inference.

**Offline Inference** We first compute the offline inference efficiency by tokenizing entire utterances. As shown in Table 6, audio codecs are significantly slower than SSL models with a similar size because the RNNs in the former cannot be parallelized in contrast to the self-attention in the latter. Next, NAST models are slow because the architecture is a Conformer encoder stacked on top of a HuBERT model (Gulati et al., 2020).[8] HuBERT Base with DC-Spin and K-means have the same inference speed since the encoder and the quantization operation are similar. In addition, large HuBERT models (300M+ parameters) are slow, which is less ideal for real-time speech tokenization.

---

[7] https://github.com/ShovalMessica/NAST/tree/main/augmentations

[8] NAST$_{50}$ and NAST$_{100}$ have the same model architecture and size.

Table 6: Offline speech tokenizer inference efficiency. Only the first RVQ codebook in the audio codec models is included in the parameter calculation.

| Method | Params | Latency↓ | RTF↓ |
|---|---|---|---|
| EnCodec (Défossez et al., 2023) | 77M | 51 ms | 0.007 |
| SpeechTokenizer (Zhang et al., 2024) | 70M | 58 ms | 0.008 |
| NAST$_{50}$ (Messica & Adi, 2024) | 220M | 64 ms | 0.009 |
| NAST$_{200}$ (Messica & Adi, 2024) | 179M | 51 ms | 0.008 |
| HuBERT Base + K-means | 95M | **18 ms** | **0.003** |
| HuBERT Large + K-means | 317M | 27 ms | 0.004 |
| HuBERT X-Large + K-means | 964M | 60 ms | 0.009 |
| HuBERT Base + DC-Spin (**proposed**) | 96M | 19 ms | **0.003** |

Table 7: Chunk-wise streaming speech tokenizer inference efficiency with 500 units, $T_{\text{chunk}} = 1$ and $T_{\text{shift}} = 0.4$ sec.

| Method | Average Latency↓ | UED↓ | TSC↑ | Resynthesis ASR-WER↓ |
|---|---|---|---|---|
| **Offline** | | | | |
| K-means | 18 ms | 0 | 63.2 | 24.2 |
| DC-Spin | 20 ms | 0 | 67.5 | 21.9 |
| **Streaming** | | | | |
| K-means | 19 ms | 11.8 | 62.8 | 25.1 |
| DC-Spin | 16 ms | 9.9 | 67.6 | 23.7 |

Table 8: SSL pre-trained encoders comparison. Unless specified otherwise, discrete units are K-means clustered hidden features with 500 centroids. We report the sWUGGY in-vocab subset, the LibriSpeech test-other for SLM ASR, and the test set for resynthesis.

| Method | Params | Pre-train Data (hours) | Spoken LM | | | | Resynthesis |
|---|---|---|---|---|---|---|---|
| | | | TSC↑ | sWUGGY↑ | sBLIMP↑ | ASR↓ | ASR-WER↓ |
| **HuBERT it2 (Hsu et al., 2021)** | | | | | | | |
| Base@50Hz | 95M | 960 | 63.2 | 74.7 | 55.5 | 18.2 | 24.4 |
| + DC-Spin$_{500,4096}$ | 96M | 960 | 67.5 | 81.4 | 60.8 | 12.2 | 22.4 |
| Large@50Hz | 317M | 60k | 66.1 | 59.7 | 56.7 | 14.7 | 25.5 |
| X-Large@50Hz | 964M | 60k | 64.5 | 75.5 | 56.3 | 13.5 | 20.9 |
| **HuBERT it3** | | | | | | | |
| Base@50Hz | 95M | 124k | 66.4 | 71.9 | 57.1 | 15.1 | 22.0 |
| Base@25Hz | 95M | 124k | 67.0 | 77.0 | 57.4 | 13.5 | 25.1 |
| Base@12.5Hz | 95M | 124k | 63.9 | 72.7 | 57.2 | 26.7 | 53.3 |
| Base@50Hz 6-Layer | 52M | 124k | 66.3 | 68.0 | 55.3 | 17.4 | 23.1 |
| Base@50Hz 18-Layer | 137M | 124k | 66.1 | 74.5 | 57.2 | 14.5 | 20.9 |
| **SpinHuBERT (proposed)** | | | | | | | |
| Base@50Hz | 95M | 124k | 67.8 | 79.4 | 59.3 | 12.1 | 21.2 |
| + DC-Spin$_{500,4096}$ | 96M | 124k | 70.7 | 82.2 | 62.8 | 11.1 | 21.7 |
| Base@25Hz | 95M | 124k | 69.6 | 78.5 | 61.0 | 11.1 | 25.5 |

**Chunk-wise Streaming**   To optimize user experience with SLMs, we repurpose speech tokenizers by chunk-wise token extraction to simulate streaming tokenization. Initially, a tokenizer extracts the first chunk of speech with a duration of $T_{\text{chunk}}$ seconds. And each time, the chunk expands by $T_{\text{shift}}$ seconds to tokenize the incoming audio. Hence, the context is constantly expanding to improve tokenization accuracy. As shown in Table 7, the proposed DC-Spin has less performance degradation than K-means and maintains downstream performance like TSC. The results demonstrate the feasibility of repurposing to streaming mode without re-training. Combining the offline and streaming experiments, DC-Spin satisfies the fourth qualification of being a good speech tokenizer. Appendix G has a more detailed explanation of the chunk-wise approach and additional results.

## 4.6   EFFECTS OF SSL PRE-TRAINING

To understand the effects of SSL pre-training on tokenizing speech, we train HuBERT models with different sizes and objectives, quantize hidden representations with K-means for speech tokenization, and report the results in Table 8. SLM ASR is the result of pre-trained SLM fine-tuned with ASR transcription (see Appendix A.5).

First, HuBERT second iteration (it2) models perform similarly on several SLM tasks, but HuBERT Large exhibits significantly worse accuracy on sWUGGY, the cause of which remains unknown even though we trained the SLM twice to verify. The results suggest scaling model size helps SLM-ASR and resynthesis but is not always helpful and also decreases inference efficiency (Table 6). Second, we pre-train HuBERT it3 models with different framerates and sizes. Compared to different framerates, 25Hz offers the best overall SLM results, but resynthesis intelligibility is degraded because the lowered framerate increases reconstruction difficulty. Like HuBERT it2, we found improvement for all metrics when scaling the model size (6 vs. 12 vs. 18 layers). Third, SpinHuBERT surpasses Hu-

BERT it3 on all tasks, indicating that enhancing pseudo labels for pre-training has a greater impact on performance than scaling the model. SpinHuBERT even narrows the performance gap between 50 and 25Hz models. Comparing K-means with DC-Spin (gray fonts), the performance gain from applying DC-Spin is more significant than all other effects. Thus, results suggest we should focus more on tokenization techniques than scaling SSL encoders.

### 4.7  FINDING PROXY TASKS FOR SPOKEN LANGUAGE MODELING

This section inspects the correlation between tasks to find proper proxies for the actual SLM tasks. See Appendix A.7 for more details.

**Bitrate**  We compute the bitrate of deduplicated tokens by considering the distribution of tokens via entropy.

**N-gram Predictability**  We propose training a 4-gram LM with deduplicated tokens on LibriSpeech and reporting the average perplexity. This metric measures the difficulty of modeling speech tokens.

**Phonetic ABX**  ABX error rate quantifies how well a tokenizer can distinguish phonemes (Schatz, 2016; Nguyen et al., 2020).

**Phone Normalized Mutual Information (PNMI)**  Proposed by Hsu et al. (2021), PNMI computes the mutual information between the speech tokens and phoneme alignments. Thereby, higher values imply better alignment with the underlying phoneme distribution.

**Character Normalized Mutual Information (CNMI)**  Similar to PNMI, CNMI compares tokens with character alignments (Chang & Glass, 2024). We use UnitY2 to compute alignments (Seamless Communication et al., 2023).[9]

Figure 3: Pearson correlation coefficients between proxy and downstream tasks.

Using 33 tokenizers with 50Hz framerate and 500 units, we compute the Pearson correlation coefficients between proxy and downstream metrics in Figure 3. We make the values negative before calculating the coefficients for lower-better metrics (bitrate, 4-gram, ABX, ASR, and resynthesis). According to Figure 3, bitrate positively correlates with TSC and sBLIMP, implying short and compact tokens are more suitable for capturing the long context of speech. Next, low 4-gram perplexity correlates with SLM tasks, so repeating patterns in tokens improves SLM. The high correlation between PNMI, ABX, and sWUGGY verifies that sWUGGY relies on well-aligned phonetic units (Section 4.2). Similarly, CNMI quantifies the textual alignment quality, making this task more related to sBLIMP and ASR. Nevertheless, the ABX error rate negatively correlates with TSC and sBLIMP, implying this metric might fail to serve as a proxy. Furthermore, speech resynthesis highly correlates with phoneme alignment metrics (ABX and PNMI), suggesting this task relies on the phonetic representations captured by the tokens for synthesizing intelligible speech signals. Overall, n-gram predictability, PNMI, and CNMI are ideal proxies for developing speech tokenizers. More results can be found in Appendix H.

## 5  CONCLUSION

This paper studies building and evaluating effective and robust speech tokenizers for spoken language modeling and speech resynthesis. We propose SpinHuBERT and DC-Spin, which demonstrate strong capabilities on several tasks compared with open-source speech tokenizers. Our methods satisfy the four qualifications for an ideal tokenizer: captures phonetic information, preserves acoustic details for resynthesis, is robust to perturbations, and fast inference. Furthermore, we found n-gram predictability, PNMI, and CNMI metrics highly correlate with downstream performance, making these tasks ideal proxies. The findings and proxy tasks offer guidelines for future tokenizer and spoken language model development.

**Limitations and Future Works**  This paper focuses on the effectiveness of speech tokenizers, so the evaluation tasks are on a smaller scale. Although the proposed tokenizers achieve state-of-the-art zero-shot metrics with small SLMs, it is worth investigating their gains on multimodal LLMs. Our models are trained and evaluated on English speech, so extending to multilingual and general audio is left for future studies. TTS and speech-to-speech translation are also potential applications.

---

[9]`https://github.com/facebookresearch/seamless_communication/blob/main/docs/m4t/unity2_aligner_README.md`

ETHICS STATEMENT

The speech tokenizers in this paper are trained with a limited set of audio data from several English corpora, which is inherently biased toward specific accents and dialects and might be less robust to unseen acoustic domains. Because inaccurate tokens might lead to misinterpretation in spoken language models, the proposed tokenizers must be carefully examined when they are used for speech processing applications.

REPRODUCIBILITY STATEMENT

The experiments of this paper utilize publicly available datasets and code for better reproducibility. First, we use public datasets for model training and evaluation as described in Section 4.1 and Appendix A. Second, the baseline speech tokenizers and SSL speech encoders are open models that can be accessed easily, as listed in Appendix A.1. Third, the training code of Spin and DC-Spin is first adopted from the official code in Chang et al. (2023) and reimplemented in the open-source fairseq library (Ott et al., 2019). We also demonstrate that our implementation matches the original performance in Appendix A.4. Fourth, we follow the original implementation for the evaluation tasks to ensure a fair comparison with prior works. For reference, we provide the source of the code and data we use in footnotes throughout the paper.

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

# A IMPLEMENTATION DETAILS

## A.1 BASELINES

**K-means** Following Hsu et al. (2021), we train the K-means models with 100 hours of speech from LibriSpeech (Panayotov et al., 2015). Table 9 lists the K-means clustering setup for the encoders we use in this paper, especially in Table 8.

**EnCodec** We use the EnCodec 24kHz model trained on speech and general audio (Défossez et al., 2023).[10] This model consists of an encoder, a residual vector quantizer (RVQ), and a decoder. We use the codeword IDs extracted from the first codebook for SLM-related tasks. Note that the speech tokens have a framerate of 75Hz. Because this model takes audio input at 24kHz, we upsample audio to 24kHz before feeding it into the encoder and downsample the decoder output to 16kHz.

**SpeechTokenizer** We adopt the official checkpoint trained on LibriSpeech (Zhang et al., 2024).[11] The architecture is similar to EnCodec, but the framerate of speech tokens is 50Hz.

**Noise Aware Speech Tokenization (NAST)** We follow the official implementation and checkpoints for NAST (Messica & Adi, 2024).[12]. However, we found the inference function applies Gumbel noise before computing the probability distribution over the codewords (Jang et al., 2017), leading to random output tokens and degrading the performance. Hence, we skip the Gumbel Softmax and residual information computation for accurate token prediction and faster inference.

Table 9: Layers for K-means clustering and the corresponding token quality in PNMI and CNMI with $K = 500$.

| Model | K-means Layer | PNMI | CNMI |
|---|---|---|---|
| **HuBERT** | | | |
| Base | 9 | 0.658 | 0.561 |
| Large | 24 | 0.670 | 0.571 |
| X-Large | 48 | 0.664 | 0.567 |
| **HuBERT it3** | | | |
| Base@50Hz | 12 | 0.669 | 0.568 |
| Base@25Hz | 11 | 0.664 | 0.561 |
| Base@12.5Hz | 7 | 0.603 | 0.477 |
| Base@50Hz 6-Layer | 6 | 0.659 | 0.562 |
| Base@50Hz 18-Layer | 18 | 0.670 | 0.570 |
| **SpinHuBERT** | | | |
| Base@50Hz | 12 | **0.688** | **0.593** |
| Base@25Hz | 12 | 0.680 | 0.584 |
| **Whisper** | | | |
| Small | 9 | 0.624 | 0.531 |

## A.2 SELF-SUPERVISED PRE-TRAINING

**LibriSpeech** LibriSpeech is a labeled read English speech corpus commonly used for ASR and SSL pre-training (Panayotov et al., 2015). The training set comprises 960 hours of speech. The four evaluation subsets are used in this paper: dev-clean, dev-other, test-clean, test-other. The Base speech SSL models from prior works are pre-trained with the 960 hours training set, including wav2vec 2.0 (Baevski et al., 2020), HuBERT (Hsu et al., 2021), data2vec (Baevski et al., 2022), and DinoSR (Liu et al., 2023).

**English Unlabeled 124k** To improve robustness, we pre-train HuBERT models with a larger English corpus, covering more audio domains. The 124k hours unlabeled speech corpus combines the English subsets Common Voice (Ardila et al., 2020), Fisher (Cieri et al., 2004), Multilingual LibriSpeech (Pratap et al., 2020), Voxlingua (Valk & Alumäe, 2021), VoxPopuli (Wang et al., 2021), LibriSpeech, and a subset originating from a publicly available repository of crawled web data. Different from prior works, we exclude Libri-Light (Kahn et al., 2020) because this corpus slightly degrades performance on domains other than LibriSpeech.

HuBERT it3 and SpinHuBERT models are pre-trained on the 124k hours dataset using 32 NVIDIA 80GB A100 GPUs with the same hyperparameters as the HuBERT Base iteration 2 (Hsu et al., 2021).[13] The only difference is that we increase the batch size to 225 seconds of speech per GPU or, equivalently, two hours considering all 32 GPUs. We list the differences for models operating at different framerates in Table 10.

---

[10] https://huggingface.co/facebook/encodec_24khz
[11] https://github.com/ZhangXInFD/SpeechTokenizer
[12] https://github.com/ShovalMessica/NAST
[13] https://github.com/facebookresearch/fairseq/tree/main/examples/hubert

Table 10: HuBERT pre-training hyperparameters for models operating at different framerates.

| Hyperparameters | 50Hz | 25Hz | 12.5Hz |
|---|---|---|---|
| CNN Extractor Layers | 7 | 8 | 9 |
| CNN Positional Encoding Kernel | 128 | 64 | 32 |
| Time Mask Length (frames) | 10 | 5 | 2 |

Additionally, the targets for SpinHuBERT Base@50Hz are derived from a $Spin_{4096}$ model based on HuBERT Base and fine-tuned with LibriSpeech 960 hours dataset, which is the same setup as will be described in Appendix A.4. The 25Hz targets are generated by another $Spin_{4096}$ model, but this model downsamples the encoder representations by averaging every two consecutive frames before performing online clustering.

## A.3 ASR & PR FINE-TUNING

**English Labeled 3k** This 3k-hour labeled English speech corpus extends from LibriSpeech by including the transcribed English subsets in Common Voice (Ardila et al., 2020) and VoxPopuli (Wang et al., 2021), filtered with the same recipe in Seamless Communication et al. (2023). We normalize the transcriptions to match LibriSpeech, i.e., removing all punctuation except for apostrophes and converting numbers to words. E.g., converting "16th" to "SIXTEENTH." After normalization and removing ambiguous transcriptions, the corpus has 3100 hours of speech left. All ASR experiments are character-based. For phonemized transcription, we use the official LibriSpeech lexicon to convert English words into phonemes.[14] We take the first pronunciation for words with multiple pronunciations for a deterministic behavior. For out-of-vocabulary words, we use a neural network-based G2P model to obtain the phoneme transcriptions (Park & Kim, 2019).

The ASR and PR fine-tuning hyperparameters are shown in Table 11. The ASR and PR models are trained on 8 NVIDIA 32GB V100 GPUs using CTC loss (Graves et al., 2006). For the first 10k updates, the encoder is frozen, and the linear projector for CTC is fine-tuned. Note that the hyperparameters are not tuned to optimal, so there still might be room for improvement.

Table 11: Hyperparameters for ASR and PR fine-tuning.

| Dataset (hours) | Training Updates | Batch Size (minutes) | Learning Rate | Time Mask Probability |
|---|---|---|---|---|
| 1 | 13k | 60.0 | 5e-5 | 0.075 |
| 10 | 25k | 26.7 | 2e-5 | 0.075 |
| 100 | 80k | 26.7 | 3e-5 | 0.075 |
| 960 | 150k | 26.7 | 3e-5 | 0.065 |
| 3100 | 150k | 26.7 | 3e-5 | 0.065 |

## A.4 SPIN & DC-SPIN

Following Chang et al. (2023), we reimplement Spin with fairseq for better scalability (Ott et al., 2019).[15] All Spin models, including DC-Spin, are trained on a single NVIDIA 32GB V100 GPU for 20k updates. We perturb each utterance in the LibriSpeech dataset with the same implementation in Chang et al. (2023) before training to avoid on-the-fly data augmentation and reduce costs. The learning rate linearly ramps up to 5e-5 in the first 4k updates and decreases to zero in the rest. The batch size is 400 seconds of audio before speaker perturbation, equivalent to 20k frames for 50Hz models. As shown in Table 12, we found that adding a small portion of masking improves performance slightly, so masking with a probability of 0.01 and a span of 5 frames is added to the input. Predicting hard targets (one-hot) also offers better alignment with phonemes and characters. Because we only fine-tune the Base models with 12 transformer encoder layers, all Spin and DC-Spin models freeze the first nine layers and fine-tune the last three layers with the learnable codebook(s). The codeword embedding size is set to 256.

---

[14] http://www.openslr.org/11/
[15] https://github.com/vectominist/spin

Table 12: DC-Spin$_{50,4096}$ with different training strategies. Soft target uses the probability distribution derived through the Sinkhorn-Knopp algorithm, while hard target converts the distribution to one-hot by taking argmax over all possible codewords.

| Spin Swapped Prediction Target | Masking | PNMI | CNMI |
|---|---|---|---|
| Hard | N/A | 0.485 | 0.350 |
| Soft | $p = 0.01$ and length $= 5$ | 0.482 | 0.349 |
| Hard | $p = 0.01$ and length $= 5$ | **0.490** | **0.355** |

Table 13: Codebook quality comparison between the original Spin implementation in Chang et al. (2023) and ours. All models are based on HuBERT Base.

| Method | Cluster Purity | Phone Purity | PNMI |
|---|---|---|---|
| Spin$_{500}$ (original) | 0.085 | 0.693 | 0.707 |
| Spin$_{500}$ (ours) | 0.082 | 0.687 | 0.702 |
| Spin$_{1000}$ (original) | 0.047 | 0.732 | 0.747 |
| Spin$_{1000}$ (ours) | 0.049 | 0.721 | 0.741 |
| Spin$_{2000}$ (original) | 0.027 | 0.757 | 0.774 |
| Spin$_{2000}$ (ours) | 0.026 | 0.759 | 0.777 |

Furthermore, to ensure that the reimplemented Spin in fairseq has a similar performance as in Chang et al. (2023), we report a comparison of Spin codebook quality between the original and our implementation in Table 13. Cluster purity measures the purity of each phoneme's associated token, and phone purity measures the average phoneme purity within one class of tokens (Hsu et al., 2021). The small discrepancy in codebook quality metrics indicates our implementation successfully reproduces the results in Chang et al. (2023).

### A.5 SPOKEN LANGUAGE MODEL

**Pre-training** Following prior works (Lakhotia et al., 2021; Gat et al., 2023; Messica & Adi, 2024), we pre-train unit-based SLMs with speech tokens extracted from the 6k hours clean subset of Libri-Light corpus (Kahn et al., 2020).[16] We select 1% of the training data for validation, which covers all sequence lengths. The unit-based LM has the `transformer_lm_big` architecture implemented in fairseq. The LM is trained on 8 NVIDIA 32GB V100 GPUs with a gradient accumulation of 8 steps, a maximum of 8192 tokens per GPU, and 3072 tokens per utterance. Utterances with lengths exceeding 3072 tokens are split into shorter sequences. The learning rate linearly increases to 5e-4 in the first 4k steps and decays as in Vaswani et al. (2017). We choose the checkpoint with the lowest validation perplexity for zero-shot evaluation and ASR fine-tuning.

**ASR Fine-tuning** Similar to SLM pre-training, ASR fine-tuning has the same training setup except for the data preparation. We extract tokens and concatenate transcription from the LibriSpeech 960h training corpus. To construct the ASR data, a special token `<|asr|>` is inserted after each tokenized utterance, followed by the corresponding character-based transcription. E.g., an utterance for training would look like `69 10 ... 11 482 <|asr|> Z E U S | S E E S | ...`, where "|" denotes whitespace. The LM input embedding is extended by randomly initializing embeddings for English letters and the special `<|asr|>` token. The training batch size and computing resources are the same as SLM pre-training. The learning rate linearly increases to 2e-4 in the first 2k steps and decays as in Vaswani et al. (2017). After training, we decode with a beam size of 5 using the checkpoint at 10k updates.

---

[16]`https://github.com/facebookresearch/fairseq/tree/main/examples/textless_nlp/gslm/ulm`

### A.6 Speech Resynthesis

**Expresso**    Expresso is a high-quality expressive speech dataset covering 26 expressive styles (Nguyen et al., 2023). This dataset is split into train, dev, and test sets. We use the train set to train a Hifi-GAN vocoder. During evaluation, we resynthesize speech in dev and test sets with the vocoder conditioned on the original speaker and style IDs. Note that some expressive styles in this dataset differ from normal speech, e.g., whispering, leading to a lower UTMOS.

We follow the default training hyperparameters in Nguyen et al. (2023) to train Hifi-GAN models on a single NVIDIA 32GB V100 GPU for 400k updates.[17] We add extra upsample layers in the Hifi-GAN for lower framerate models like 25Hz and 12.5Hz.

### A.7 Proxy Task Metrics

**Bitrate**    We consider the distribution of the units extracted by the tokenizers during bitrate calculation. Assuming a corpus of $T$ seconds of audio and $N$ tokens in total. For each token ID $k = 1, \ldots, K$, the number of occurrences is denoted as $n(k)$. Hence, the probability of occurrence of each token $k$ is $p(k) = n(k)/N$. Then, we calculate the bitrate as follows

$$
\begin{aligned}
\text{bitrate} &= \frac{N}{T} \mathbb{E}\left[-\log_2 p(k)\right] \\
&= \frac{N}{T} \sum_{k=1}^{K} \frac{n(k)}{N} \log_2 \frac{N}{n(k)} \\
&\leq \frac{N}{T} \log_2 K.
\end{aligned}
$$

The bitrates are calculated over the dev-clean and dev-other subsets of LibriSpeech.

**N-gram Predictability**    We implement a simple n-gram LM and estimate unseen n-grams by backing off to lower-order n-gram LMs. First, all audio data in LibriSpeech are tokenized and deduplicated. Second, for each utterance, we add `<|bos|>` and `<|eos|>` tokens at the front and end, respectively. Then, we train n-gram LMs with orders from $1, \ldots, n$. These LMs then estimate the log probability of the dev and test sets in LibriSpeech to get the perplexities. Lower perplexities indicate that the tokens are easier to be predicted given a small context, which also implies similar token patterns appear frequently.

---

[17]`https://github.com/facebookresearch/speech-resynthesis/tree/main/examples/expresso`

## B  DC-Spin: Design and Analysis

### B.1  Effect of the Auxiliary Codebook

Here, we discuss the effect of the auxiliary codebook size in DC-Spin. We plot the relation between the auxiliary codebook sizes and the zero-shot SLM results in Figure 4. Comparing Spin and DC-Spin (dashed vs. solid lines), the proposed DC-Spin helps downstream tasks in most cases. Moreover, the performance gain of larger auxiliary codebook sizes is more prominent in sWUGGY than in the other two tasks, corroborating with the findings in Chang et al. (2023) and the discussions in Section 4.2. Still, we observe that the overall performance drops when the codebook size is over 4096. The results indicate the necessity of including a large auxiliary codebook for helping the primary Spin codebook for SLM applications.

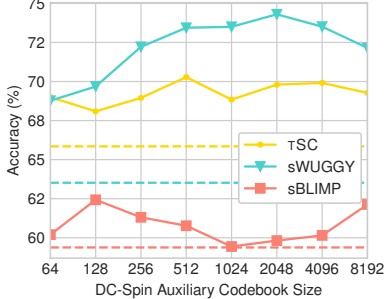

Figure 4: DC-Spin$_{50,\cdot}$ with different auxiliary codebook sizes vs. zero-shot SLM tasks. Dashed lines indicate Spin$_{50}$.

### B.2  Quantization: K-means vs. Spin Codebook

Here, we discuss the difference between quantizing speech encoder representations with Spin codebook and K-means clustering. Among K-means results in Table 14, Spin$_{4096}$ offers the overall best performance because the large codebook used during self-supervised fine-tuning enhances the encoder in capturing phonetic units. Still, the gap between Spin$_{4096}$ and DC-Spin is narrowed when K-means has 500 centroids. When comparing the two quantization techniques, K-means vs. codebook, we found that the codebook quantization method is slightly better because the codebooks are optimized jointly with the encoder. The results demonstrate that Spin and DC-Spin codebooks are, in general, a better way of quantizing encoder embedding than K-means. Another benefit of using Spin codebooks is to avoid the need for training a separate K-means model.

Table 14: Zero-shot SLM results with different quantization approaches. All models are based on HuBERT Base. K-means clustering is performed on the transformer encoder output.

| Units | Fine-tuning | Quantization | PNMI↑ | τSC↑ | sWUGGY↑ all | in-vocab | sBLIMP↑ |
|-------|-------------|--------------|-------|------|-------------|----------|---------|
| 50    | Spin$_{50}$ | K-means | 0.481 | 65.42 | 57.92 | 62.48 | 58.46 |
|       | Spin$_{4096}$ | K-means | 0.481 | 68.15 | **67.44** | **76.27** | 59.89 |
|       | DC-Spin$_{50,4096}$ | K-means | **0.496** | 69.05 | 64.14 | 71.69 | 59.34 |
|       | Spin$_{50}$ | Codebook | 0.482 | 65.85 | 58.90 | 63.52 | 59.38 |
|       | DC-Spin$_{50,4096}$ | Codebook | 0.490 | **69.91** | 65.05 | 73.51 | **60.15** |
| 100   | Spin$_{100}$ | K-means | 0.565 | 67.40 | 65.43 | 73.11 | 60.71 |
|       | Spin$_{4096}$ | K-means | **0.573** | 68.57 | **69.98** | **80.51** | 61.17 |
|       | DC-Spin$_{100,4096}$ | K-means | 0.567 | 68.36 | 68.60 | 78.44 | 61.00 |
|       | Spin$_{100}$ | Codebook | 0.565 | 68.25 | 65.28 | 73.25 | 59.97 |
|       | DC-Spin$_{100,4096}$ | Codebook | 0.558 | **70.18** | 68.04 | 78.47 | **61.35** |
| 200   | Spin$_{200}$ | K-means | 0.639 | 69.27 | 68.14 | 77.16 | **63.01** |
|       | Spin$_{4096}$ | K-means | **0.650** | 68.41 | **71.12** | **81.23** | 60.70 |
|       | DC-Spin$_{200,4096}$ | K-means | 0.641 | 69.00 | 69.88 | 79.50 | 62.42 |
|       | Spin$_{200}$ | Codebook | 0.640 | **69.64** | 68.95 | 78.19 | 62.55 |
|       | DC-Spin$_{200,4096}$ | Codebook | 0.640 | 69.21 | 70.79 | 80.59 | 62.13 |
| 500   | Spin$_{500}$ | K-means | 0.701 | 66.44 | 70.00 | 79.56 | 60.74 |
|       | Spin$_{4096}$ | K-means | 0.710 | 65.90 | 70.66 | 80.15 | 61.12 |
|       | DC-Spin$_{500,4096}$ | K-means | **0.711** | 66.44 | 70.93 | 80.44 | **61.31** |
|       | Spin$_{500}$ | Codebook | 0.702 | 67.45 | 70.03 | 79.31 | 60.08 |
|       | DC-Spin$_{500,4096}$ | Codebook | 0.709 | **67.50** | **71.48** | **81.38** | 60.84 |

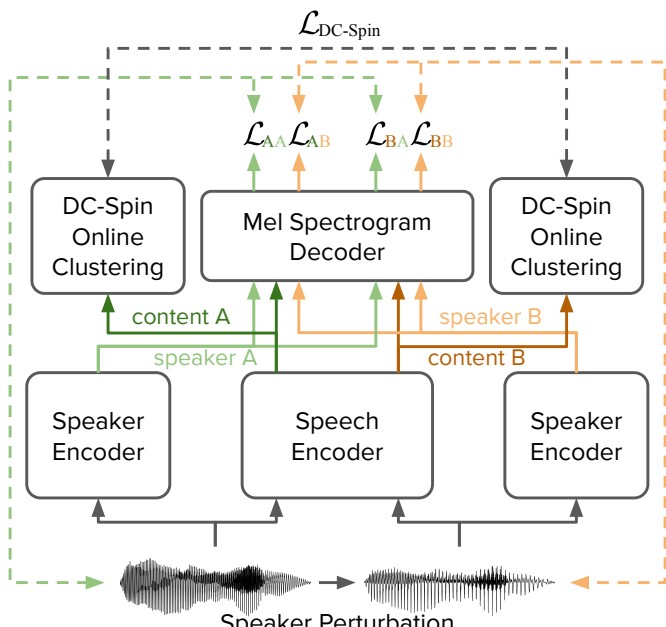

Figure 5: DC-Spin with mel spectrogram reconstruction auxiliary objective. The reconstruction loss is $\mathcal{L}_{\text{Mel}} = \mathcal{L}_{\text{AA}} + \mathcal{L}_{\text{AB}} + \mathcal{L}_{\text{BA}} + \mathcal{L}_{\text{BB}}$.

### B.3 SUPERVISED DC-SPIN FINE-TUNING

**Method** Inspired by the speech encoders in multimodal LLM studies (Gemini Team, 2023; Dubey et al., 2024), we include supervised fine-tuning to boost tokenizer quality. Different from Section 3.2, the SFT here is applied during DC-Spin fine-tuning so that the tokenizer is jointly optimized with several objectives, i.e., multitask learning. We consider three types of SFT: Mel spectrogram reconstruction (Mel), CTC-based character recognition (CTC-ASR), and CTC-based phoneme recognition (CTC-PR). Note that Mel reconstruction is unsupervised, but we consider these tasks together as SFT for a simpler presentation. For Mel reconstruction, we adopt the speaker encoder and the decoder proposed in Chou & Lee (2019). The speaker encoder transforms the Mel spectrogram of an utterance to a single speaker embedding. The decoder then reconstructs the Mel spectrogram by taking the speech encoder's output embedding and fuses the speaker embedding with adaptive instance normalization (Huang & Belongie, 2017). Because the input for Spin training consists of pairs of utterances with the same content spoken by different speakers, we reconstruct each utterance into two Mel spectrograms using the original and the perturbed speaker embeddings, as illustrated in Figure 5. This training objective is expected to help disentangle speaker and content representations. We use a standard $L^1$ loss for this task. For CTC-ASR and CTC-PR, a linear prediction is added to project hidden representations to logits over all possible output textual tokens. Thus, the loss function of the supervised DC-Spin is

$$\mathcal{L}_{\text{DC-Spin-SFT}} = \mathcal{L}_{\text{DC-Spin}} + \lambda_{\text{Mel}}\mathcal{L}_{\text{Mel}} + \lambda_{\text{CTC-ASR}}\mathcal{L}_{\text{CTC-ASR}} + \lambda_{\text{CTC-PR}}\mathcal{L}_{\text{CTC-PR}},$$

where $\lambda_{\text{Mel}}$, $\lambda_{\text{CTC-ASR}}$, and $\lambda_{\text{CTC-PR}}$ are hyperparameters. With the SFT tasks, the learned codebooks are expected to align better with the underlying textual and phonetic distribution.

**Setup** The training setup is almost identical to DC-Spin as discussed in Appendix A.4, but the peak learning rate here is 2e-5. In the experiments, we always let

$$\lambda_{\text{Mel}} + \lambda_{\text{CTC-ASR}} + \lambda_{\text{CTC-PR}} = 5$$

and assign the same value for each loss. E.g., when Mel reconstruction and PR are applied, we have $\lambda_{\text{Mel}} = \lambda_{\text{CTC-PR}} = 2.5$ and $\lambda_{\text{CTC-ASR}} = 0$. We compute 80-bin Mel spectrograms with torchaudio (Yang et al., 2022).

**Results** According to Table 15, fine-tuning DC-Spin jointly with supervised objectives offers limited improvement compared with applying ASR and PR before DC-Spin. This phenomenon might

Table 15: Supervised tokenizers on zero-shot SLM tasks. All tokenizers are DC-Spin models with auxiliary tasks indicated by the checkmarks. The top three results in each section are underlined.

| Mel | ASR | PR | τSC↑ | sWUGGY↑ all | sWUGGY↑ in-vocab | sBLIMP↑ | Mel | ASR | PR | τSC↑ | sWUGGY↑ all | sWUGGY↑ in-vocab | sBLIMP↑ |
|---|---|---|---|---|---|---|---|---|---|---|---|---|---|
| **50 units** | | | | | | | **200 units** | | | | | | |
| | | | 69.9 | 65.1 | 73.5 | 60.2 | | | | 69.2 | 70.8 | 80.6 | 62.1 |
| ✓ | | | 68.5 | 67.6 | 77.9 | 59.0 | ✓ | | | 70.0 | 71.8 | 82.0 | 61.8 |
| | ✓ | | 70.0 | 68.6 | 77.4 | 62.7 | | ✓ | | 66.6 | 70.9 | 80.4 | 61.0 |
| | | ✓ | 69.8 | 68.9 | 78.1 | 60.6 | | | ✓ | 68.4 | 71.8 | 82.3 | 60.9 |
| ✓ | ✓ | | 69.4 | 69.4 | 79.6 | 62.1 | ✓ | ✓ | | 69.9 | 69.9 | 79.5 | 60.8 |
| ✓ | | ✓ | 70.8 | 70.1 | 80.5 | 61.4 | ✓ | | ✓ | 68.8 | 71.9 | 82.1 | 60.6 |
| | ✓ | ✓ | 67.6 | 70.3 | 80.3 | 61.0 | | ✓ | ✓ | 67.5 | 71.1 | 80.6 | 61.1 |
| ✓ | ✓ | ✓ | 70.9 | 68.0 | 76.6 | 62.2 | ✓ | ✓ | ✓ | 68.3 | 72.0 | 82.2 | 59.9 |
| HuBERT-ASR$_{1k}$ | | | 70.4 | 66.5 | 75.2 | 62.7 | HuBERT-ASR$_{1k}$ | | | 70.2 | 70.4 | 80.5 | 63.6 |
| HuBERT-PR$_{1k}$ | | | 69.4 | 70.2 | 80.6 | 63.0 | HuBERT-PR$_{1k}$ | | | 69.9 | 73.3 | 83.8 | 59.6 |
| **100 units** | | | | | | | **500 units** | | | | | | |
| | | | 70.2 | 68.0 | 78.5 | 61.4 | | | | 67.5 | 71.5 | 81.4 | 60.8 |
| ✓ | | | 70.4 | 70.4 | 81.2 | 62.7 | ✓ | | | 67.6 | 70.6 | 80.3 | 60.0 |
| | ✓ | | 69.6 | 70.6 | 80.5 | 62.2 | | ✓ | | 67.2 | 70.4 | 79.9 | 59.8 |
| | | ✓ | 69.1 | 71.2 | 80.9 | 62.2 | | | ✓ | 67.0 | 72.0 | 82.0 | 58.5 |
| ✓ | ✓ | | 69.2 | 71.0 | 80.3 | 62.9 | ✓ | ✓ | | 66.5 | 70.6 | 80.0 | 59.6 |
| ✓ | | ✓ | 69.4 | 71.6 | 82.1 | 61.0 | ✓ | | ✓ | 65.6 | 70.8 | 80.6 | 58.0 |
| | ✓ | ✓ | 70.1 | 71.3 | 81.5 | 61.0 | | ✓ | ✓ | 67.8 | 71.0 | 80.4 | 59.5 |
| ✓ | ✓ | ✓ | 68.4 | 71.7 | 82.5 | 61.0 | ✓ | ✓ | ✓ | 66.1 | 72.0 | 82.3 | 59.1 |
| HuBERT-ASR$_{1k}$ | | | 72.0 | 68.9 | 78.8 | 63.5 | HuBERT-ASR$_{1k}$ | | | 69.6 | 72.4 | 82.5 | 63.5 |
| HuBERT-PR$_{1k}$ | | | 70.9 | 71.1 | 81.0 | 61.6 | HuBERT-PR$_{1k}$ | | | 69.3 | 72.3 | 82.7 | 62.5 |

be explained by the fact that ASR and PR require fine-tuning more hidden layers to perform well or have a greater impact on token quality. However, fine-tuning too many layers with Spin leads to collapsed representations and requires more advanced techniques to mitigate this issue (Chang et al., 2023; Chang & Glass, 2024), making DC-Spin + SFT more difficult to find the optimal hyperparameters. Thus, we exclude supervised DC-Spin fine-tuning from the main text and separate unsupervised and supervised fine-tuning into two stages.

## B.4 CODEBOOK QUALITY

To quantify the codebook quality, we compute the ABX, PNMI, and CNMI values for several speech tokenizers in Table 16. The ABX scores are averaged over LibriSpeech dev subsets used in ZeroSpeech 2021. For HuBERT Base tokenizers, the trend of all three metrics on K-means, Spin, and DC-Spin indicate the effectiveness of the proposed DC-Spin tokenizer in capturing better phonetic representations. However, when fine-tuned with ASR or PR, the ABX error rates increased while PNMI and CNMI were improved. We suspect this phenomenon is caused by the fact that some fine-grained phonetic representations in SSL models become coarser because of the CTC-based supervised fine-tuning tasks.

Table 16: Codebook quality of speech tokenizers with 500 units. The ABX implementation differs from Gat et al. (2023) and Messica & Adi (2024), so the scores are not comparable.

| Method | ABX↓ | PNMI↑ | CNMI↑ |
|---|---|---|---|
| **HuBERT Base** | | | |
| K-means$_{500}$ | 5.30% | 0.658 | 0.561 |
| Spin$_{500}$ | 4.48% | 0.702 | 0.585 |
| DC-Spin$_{500,4096}$ | **3.76%** | 0.709 | 0.596 |
| + ASR$_{1k}$ | 5.74% | 0.710 | **0.663** |
| + PR$_{1k}$ | 5.42% | **0.728** | 0.636 |
| **SpinHuBERT Base@50Hz** | | | |
| K-means$_{500}$ | **4.63%** | 0.688 | 0.593 |
| DC-Spin$_{500,4096}$ | 5.03% | 0.679 | 0.589 |
| + ASR$_{1k}$ | 6.60% | 0.699 | 0.648 |
| + PR$_{1k}$ | 6.47% | 0.712 | 0.622 |
| + ASR$_{3k}$ | 6.53% | 0.694 | **0.651** |
| + PR$_{3k}$ | 6.13% | **0.715** | 0.625 |

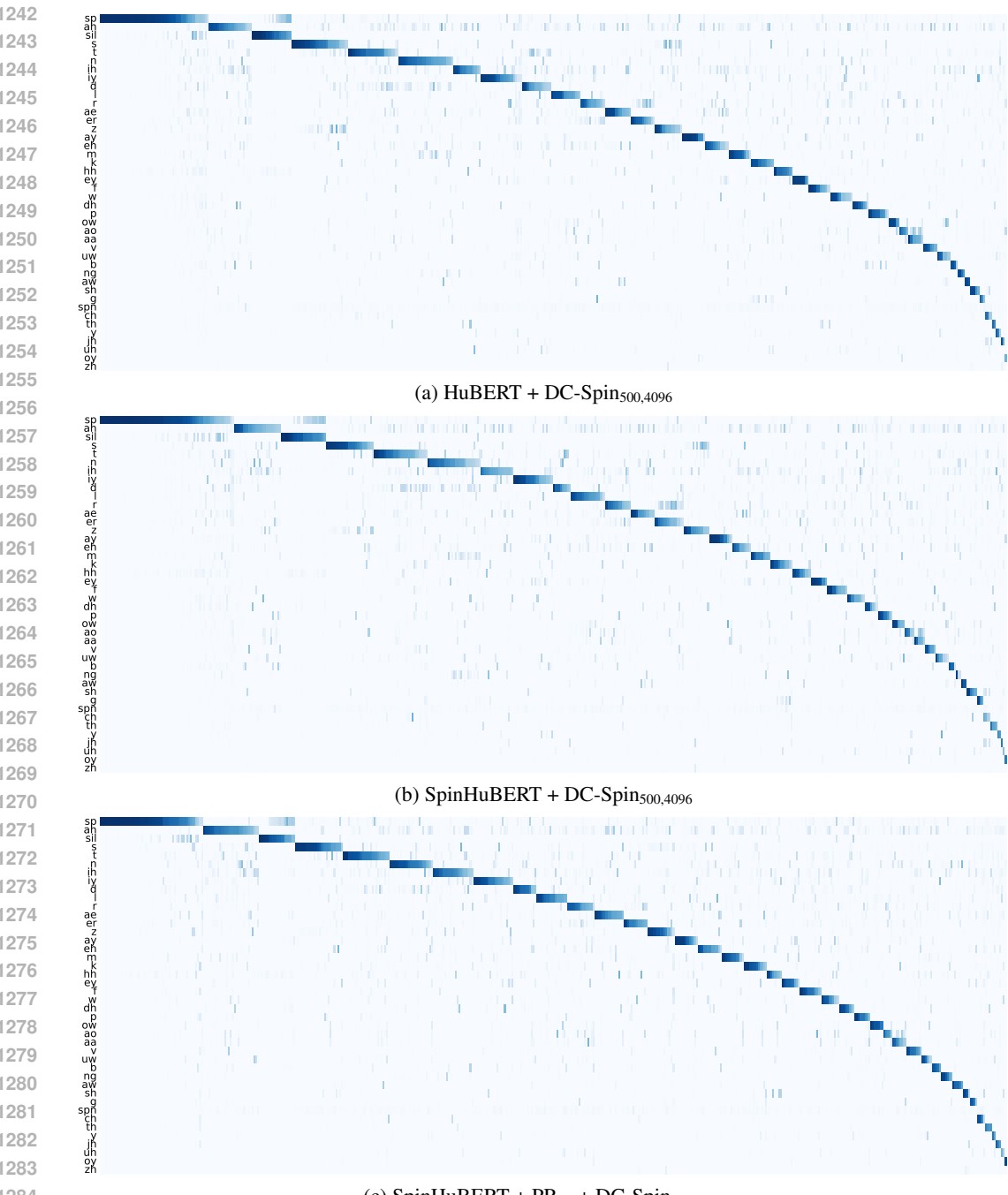

(a) HuBERT + DC-Spin$_{500,4096}$

(b) SpinHuBERT + DC-Spin$_{500,4096}$

(c) SpinHuBERT + PR$_{3k}$ + DC-Spin$_{500,4096}$

Figure 6: $P(\text{phone}|\text{code})$ visualization for different DC-Spin primary codebooks The vertical axes represent the phones sorted from high to low frequencies.

## B.5 CODEBOOK VISUALIZATION

Following prior works (Chang et al., 2023; Liu et al., 2023), we plot the $P(\text{phone}|\text{code})$ of the primary codebook in DC-Spin to visualize the assignment for each code to phonemes. Figure 6 illustrates DC-Spin with different initialization models. The distributions between different models are similar and follow the long-tail distribution of phoneme occurrences. Still, SpinHuBERT + PR fine-tuning slightly assigns the codes to each phoneme more uniformly.

## C ADDITIONAL SPOKEN LANGUAGE MODEL RESULTS

This section offers more complete results and ablation studies on SLM.

### C.1 SLM-ASR INITIALIZATION

As shown in Table 17, we compare decoder-only ASR fine-tuning with and without unsupervised SLM pre-training. The lower WERs indicate that large-scale SLM pre-training benefits downstream fine-tuning. Hence, all SLM-based ASR experiments in this paper are initialized with pre-trained SLMs.

Table 17: Decoder-only ASR WERs with and without unit-based SLM pre-training. The discrete units are HuBERT Base Layer 9 K-means 500 units.

| Method | dev-clean | dev-other | test-clean | test-other |
|---|---|---|---|---|
| From Scratch | 8.5 | 18.7 | 8.9 | 18.8 |
| SLM Pre-training | **7.7** | **17.6** | **7.9** | **18.2** |

### C.2 SLM-BASED ASR

This section reports the complete SLM-based ASR results and 4-gram predictability in Table 18. The findings are listed as follows:

1. Audio codecs offer the worst ASR WERs, showing that the first codebook poorly captures the content of speech because the information is spread out to the other RVQ codebooks.
2. Whisper encoder with K-means clustering performs worse than most SSL-based encoders. Although the Whisper encoder is trained with 680k hours of labeled speech, the sequence-to-sequence ASR architecture does not explicitly constrain the encoder to align with the input signals or capture fine-grained phonetic units, making this model less ideal for tokenizing speech than SSL models.
3. According to the HuBERT Large and X-Large results, scaling the encoders with more parameters improves ASR. However, compared with DC-Spin in the last part of the table, scaling has less impact on the performance.
4. Unsurprisingly, supervised fine-tuning with PR and ASR improves SLM-based ASR and 4-gram perplexity because the supervision directly relates to the downstream task.
5. The 4-gram predictability (perplexity) correlates with ASR WER, indicating that this metric is a good proxy for SLM-based ASR, consistent with Figure 3.

Results in Table 18 demonstrate the proposed SpinHuBERT and DC-Spin tokenizer offer the best SLM-based ASR.

Table 18: Decoder-only ASR WERs on LibriSpeech. All ASR models are initialized with pre-trained SLMs. The lowest WERs are **boldfaced**, and the second and third best values are underlined.

| Method | SLM-based ASR WER | | | | 4-gram Perplexity | | | |
| --- | --- | --- | --- | --- | --- | --- | --- | --- |
| | dev | | test | | dev | | test | |
| | clean | other | clean | other | clean | other | clean | other |
| **Audio Codecs** | | | | | | | | |
| EnCodec (Défossez et al., 2023) | 55.8 | 72.7 | 54.3 | 75.0 | 49.8 | 49.4 | 48.8 | 47.3 |
| SpeechTokenizer (Zhang et al., 2024) | 13.1 | 28.8 | 13.4 | 31.3 | 6.1 | 7.8 | 5.9 | 8.3 |
| **K-means$_{500}$** | | | | | | | | |
| Whisper (Radford et al., 2023) | | | | | | | | |
| Small | 10.7 | 22.4 | 10.5 | 22.8 | 7.9 | 9.5 | 7.9 | 9.8 |
| HuBERT (Hsu et al., 2021) | | | | | | | | |
| Base | 7.7 | 17.6 | 7.9 | 18.2 | 6.3 | 7.8 | 6.3 | 7.9 |
| + ASR$_{1k}$ | 6.1 | 11.9 | 5.7 | 11.5 | 3.8 | 4.2 | 3.8 | 4.2 |
| + PR$_{1k}$ | 5.4 | 11.4 | 5.7 | 11.5 | 4.0 | 4.5 | 3.9 | 4.5 |
| Base@25Hz♠ | 7.4 | 15.6 | 7.4 | 16.0 | 8.3 | 9.8 | 8.2 | 9.9 |
| Large | 6.5 | 14.6 | 6.4 | 14.7 | 5.5 | 6.6 | 5.5 | 6.6 |
| X-Large | 6.6 | 13.9 | 6.7 | 13.5 | 5.5 | 6.4 | 5.5 | 6.4 |
| HuBERT iteration 3 | | | | | | | | |
| Base@50Hz | 7.0 | 14.7 | 7.1 | 15.1 | 5.7 | 6.8 | 5.7 | 6.8 |
| Base@25Hz | 6.7 | 13.1 | 7.1 | 13.5 | 8.0 | 9.4 | 7.9 | 9.3 |
| Base@12.5Hz | 12.9 | 26.1 | 13.0 | 26.7 | 15.8 | 19.4 | 15.5 | 19.1 |
| Base@50Hz 6-Layer | 7.6 | 17.4 | 8.1 | 17.4 | 5.8 | 6.9 | 5.8 | 6.9 |
| Base@50Hz 18-Layer | 6.9 | 14.0 | 6.9 | 14.5 | 5.7 | 6.8 | 5.7 | 6.8 |
| SpinHuBERT | | | | | | | | |
| Base@50Hz | 6.2 | 12.0 | 6.4 | 12.1 | 3.6 | 3.9 | 3.6 | 3.9 |
| Base@25Hz | 6.6 | 11.6 | 6.5 | 11.1 | 5.5 | 5.9 | 5.3 | 5.8 |
| **Spin / DC-Spin (Proposed)** | | | | | | | | |
| HuBERT Base | | | | | | | | |
| + Spin$_{500}$ | 6.6 | 13.2 | 6.7 | 13.4 | 3.8 | 4.2 | 3.8 | 4.2 |
| + DC-Spin$_{500,4096}$ | 6.0 | 12.2 | 6.0 | 12.2 | 3.6 | 3.9 | 3.6 | 3.9 |
| + ASR$_{1k}$ | 5.2 | 10.7 | 5.7 | 10.6 | **3.2** | **3.5** | **3.2** | 3.5 |
| + PR$_{1k}$ | 5.6 | 10.8 | 5.6 | 11.2 | 3.4 | 3.6 | 3.3 | 3.6 |
| SpinHuBERT Base@50Hz | | | | | | | | |
| + DC-Spin$_{500,4096}$ | 5.9 | 10.9 | 6.0 | 11.1 | 3.7 | 3.9 | 3.7 | 3.9 |
| + ASR$_{1k}$ | 5.2 | **9.1** | **5.1** | 9.6 | 3.3 | **3.5** | 3.3 | 3.5 |
| + PR$_{1k}$ | **5.1** | 9.4 | 5.4 | 9.5 | 3.5 | 3.7 | 3.5 | 3.7 |
| + ASR$_{3k}$ | 5.3 | **9.1** | 5.4 | **9.3** | 3.3 | **3.5** | 3.3 | **3.4** |
| + PR$_{3k}$ | **5.1** | 9.2 | 5.2 | 9.6 | 3.4 | 3.6 | 3.4 | 3.6 |
| SpinHuBERT Base@25Hz | | | | | | | | |
| + DC-Spin$_{500,4096}$ | 6.4 | 10.7 | 6.5 | 10.8 | 5.2 | 5.5 | 5.1 | 5.5 |
| + ASR$_{1k}$ | 5.9 | 9.7 | 6.2 | 9.9 | 5.0 | 5.2 | 4.9 | 5.1 |
| + PR$_{1k}$ | 6.4 | 9.5 | 6.2 | 9.9 | 5.0 | 5.2 | 4.9 | 5.2 |
| + ASR$_{3k}$ | 6.0 | 9.9 | 6.3 | 10.2 | 5.0 | 5.2 | 4.9 | 5.2 |
| + PR$_{3k}$ | 6.0 | 9.8 | 6.4 | 10.0 | 4.8 | 5.1 | 4.7 | 5.0 |

♠The HuBERT model used in Nguyen et al. (2024).

## D ADDITIONAL SPEECH RESYNTHESIS RESULTS

This section reports complete results on speech resynthesis in Table 19. Note that the first two RVQ codebooks are always used during EnCodec training, so the EnCodec results start from 1.5kbps bandwidth. Because of this property, the tokens produced by the first codebook contain less useful information for downstream tasks, as shown in the zero-shot SLM experiments in Section 4.2.

In Table 19, EnCodec and SpeechTokenizer require multiple codebooks, i.e., high bitrates, to resynthesize the audio with low ASR-WER and high UTMOS. For instance, when all codebooks are used (RVQ1:8), SpeechTokenizer performs similarly to our best model but requires a 9X bitrate. The UTMOS of EnCodec is relatively low even though the intelligibility is high probably because this model is not specialized in synthesizing human speech. Next, comparing DC-Spin with different codebook sizes (the third section of Table 19), a larger primary codebook offers superior resynthesis performance because the bandwidth increases.

Extending from Section 4.3, Table 19 reports additional results using randomly selected speaker and style IDs to simulate real-world applications. Generally, we found slight degradation in random resynthesis intelligibility and similar UTMOS. Since the Spin and DC-Spin tokenizers are only trained with a speaker-invariant objective, the style information is still preserved in the tokens, making resynthesizing to a different style more difficult. One possible solution is to include more complex perturbations in the Spin fine-tuning process to force the tokenizer to neglect irrelevant information.

Table 19: Complete speech resynthesis ASR-WER and UTMOS results on Expresso dev and test sets with different methods and bitrates. "Original" and "random" respectively denote resynthesizing speech with the original and random speaker and style IDs.

| Method | Bitrate | ASR-WER↓ | | | | UTMOS↑ | | | |
| | | original | | random | | original | | random | |
| | | dev | test | dev | test | dev | test | dev | test |
| --- | --- | --- | --- | --- | --- | --- | --- | --- | --- |
| Ground Truth | 256k | 15.2 | 14.3 | – | – | 3.24 | 3.28 | – | – |
| **EnCodec (Défossez et al., 2023)** | | | | | | | | | |
| RVQ1:2 | 1.5k | 28.4 | 27.5 | – | – | 1.35 | 1.31 | – | – |
| RVQ1:4 | 3k | 19.3 | 19.3 | – | – | 1.74 | 1.67 | – | – |
| RVQ1:8 | 6k | 17.1 | 16.6 | – | – | 2.26 | 2.22 | – | – |
| RVQ1:16 | 12k | **16.4** | **16.1** | – | – | **2.65** | **2.64** | – | – |
| **SpeechTokenizer (Zhang et al., 2024)** | | | | | | | | | |
| RVQ1 | 500 | 30.7 | 32.9 | – | – | 1.27 | 1.27 | – | – |
| RVQ1:2 | 1k | 25.4 | 25.2 | – | – | 2.25 | 2.00 | – | – |
| RVQ1:4 | 2k | 20.7 | 20.5 | – | – | 2.76 | 2.63 | – | – |
| RVQ1:8 | 4k | **18.8** | **18.4** | – | – | **2.94** | **2.91** | – | – |
| **HuBERT** | | | | | | | | | |
| K-means$_{500}$ | 448 | 24.0 | 24.4 | 26.0 | 25.3 | 2.93 | 2.76 | 2.92 | 2.91 |
| DC-Spin$_{50,4096}$ | 282 | 33.3 | 33.9 | 38.7 | 39.2 | 2.89 | 2.80 | 2.79 | 2.79 |
| DC-Spin$_{100,4096}$ | 332 | 26.9 | 27.6 | 29.6 | 29.8 | 2.99 | 2.91 | 2.93 | 2.93 |
| DC-Spin$_{200,4096}$ | 382 | 22.8 | 25.2 | 25.9 | 26.9 | 2.89 | 2.73 | 2.82 | 2.84 |
| DC-Spin$_{500,4096}$ | 448 | **21.3** | **22.4** | **23.4** | **24.2** | 2.96 | 2.93 | 2.92 | 2.93 |
| + ASR$_{1k}$ | 448 | 21.6 | 22.9 | 23.8 | 25.1 | 2.96 | 2.96 | 2.89 | 2.89 |
| + PR$_{1k}$ | 448 | 21.4 | 22.5 | 23.5 | 24.4 | **3.00** | **2.97** | **2.99** | **2.98** |
| **SpinHuBERT Base@50Hz** | | | | | | | | | |
| K-means$_{500}$ | 448 | 20.0 | 21.2 | 21.5 | 22.4 | 3.05 | 2.94 | 2.98 | 2.99 |
| DC-Spin$_{500,4096}$ | 448 | 20.5 | 21.7 | 22.5 | 23.2 | **3.11** | 3.04 | **3.00** | **3.00** |
| + ASR$_{1k}$ | 448 | 21.7 | 22.6 | 24.2 | 24.3 | 2.90 | 2.84 | 2.86 | 2.87 |
| + PR$_{1k}$ | 448 | 21.0 | 20.7 | 24.6 | 24.1 | 2.93 | 2.84 | 2.88 | 2.88 |
| + ASR$_{3k}$ | 448 | 18.9 | 20.0 | 23.2 | 23.7 | 3.08 | **3.05** | 2.98 | 2.99 |
| + PR$_{3k}$ | 448 | **18.8** | **18.7** | **21.6** | **21.3** | 3.02 | 2.92 | 2.97 | 2.97 |

# E  SELF-SUPERVISED PRE-TRAINING

This section reports and discusses additional results of SSL pre-training, including ASR fine-tuning, SUPERB downstream evaluation, and layer-wise analysis.

## E.1  CTC-BASED AUTOMATIC SPEECH RECOGNITION

Following prior studies, we fine-tune SSL models with limited labeled data for ASR with the setup described in Appendix A.3 (Baevski et al., 2020; Hsu et al., 2021; Chen et al., 2022). The experiments here are conducted once without hyperparameter tuning, which might not reflect the true performance of SpinHuBERT. As shown in Table 20, SpinHuBERT outperforms HuBERT it3 in all setups, showing that improving HuBERT pre-training targets helps capture better content information and offers a better initialization for ASR fine-tuning. However, HuBERT it3 is slightly worse than HuBERT it2 (Hsu et al., 2021), which might be caused by the fact that HuBERT it3 is trained with a more diverse and noisy dataset without a denoising objective like WavLM (Chen et al., 2022), while the training and evaluation of HuBERT it2 are both on the clean LibriSpeech corpus. Moreover, the HuBERT it3 and SpinHuBERT models are trained with 124k hours of speech but are optimized with only 400k steps, significantly fewer than that of WavLM Base+ (1M steps). Although SpinHuBERT is slightly inferior to the prior state-of-the-art like multi-resolution HuBERT (Shi et al., 2024a) and WavLM in some ASR cases, the main purpose of developing SpinHuBERT is to offer a better initialization for the proposed DC-Spin.

Table 20: LibriSpeech CTC-based ASR results without LM.

| Model | Pre-train Data (hours) | dev clean | dev other | test clean | test other |
|---|---|---|---|---|---|
| **1h labeled** | | | | | |
| wav2vec 2.0 Base (Baevski et al., 2020) | 960 | 24.1 | 29.6 | 24.5 | 29.7 |
| HuBERT Base (Hsu et al., 2021)♠ | 960 | 20.2 | 28.1 | 20.6 | 28.9 |
| WavLM Base (Baevski et al., 2022) | 960 | – | – | 24.5 | 29.2 |
| WavLM Base+ (Chen et al., 2022) | 94k | – | – | 22.8 | 26.7 |
| MR-HuBERT mono-base (Shi et al., 2024a) | 960 | **18.8** | **23.7** | **19.3** | 24.5 |
| HuBERT it3 Base@50Hz | 124k | 22.4 | 28.1 | 22.2 | 28.3 |
| SpinHuBERT Base@50Hz | 124k | 19.6 | 24.4 | 19.7 | **24.4** |
| **10h labeled** | | | | | |
| wav2vec 2.0 Base (Baevski et al., 2020) | 960 | 10.9 | 17.4 | 11.1 | 17.6 |
| HuBERT Base (Hsu et al., 2021)♠ | 960 | 9.6 | 16.6 | 9.7 | 17.0 |
| WavLM Base (Baevski et al., 2022) | 960 | – | – | 9.8 | 16.0 |
| WavLM Base+ (Chen et al., 2022) | 94k | – | – | 9.0 | 14.7 |
| MR-HuBERT mono-base (Shi et al., 2024a) | 960 | **8.5** | **13.2** | **8.5** | **13.5** |
| HuBERT it3 Base@50Hz | 124k | 10.7 | 17.1 | 10.6 | 17.4 |
| SpinHuBERT Base@50Hz | 124k | 9.3 | 14.7 | 9.3 | 14.7 |
| **100h labeled** | | | | | |
| wav2vec 2.0 Base (Baevski et al., 2020) | 960 | 6.1 | 13.5 | 6.1 | 13.3 |
| HuBERT Base (Hsu et al., 2021)♠ | 960 | 5.8 | 12.9 | 5.8 | 12.8 |
| WavLM Base (Baevski et al., 2022) | 960 | – | – | 5.7 | 12.0 |
| WavLM Base+ (Chen et al., 2022) | 94k | – | – | 4.6 | 10.1 |
| MR-HuBERT mono-base (Shi et al., 2024a) | 960 | 4.9 | **9.0** | 4.9 | **9.2** |
| HuBERT it3 Base@50Hz | 124k | 5.5 | 12.0 | 5.6 | 12.1 |
| SpinHuBERT Base@50Hz | 124k | **4.8** | 10.6 | **4.8** | 10.4 |

♠Source: Shi et al. (2024a)

Table 21: Results of SSL models on SUPERB. "ParaL." indicates the paralinguistic task. Unless specified otherwise, all models are "Base" with approximately 95M parameters.

| | Content | | | | Semantics | | | | ParaL. | Speaker | | |
| --- | --- | --- | --- | --- | --- | --- | --- | --- | --- | --- | --- | --- |
| | PR | ASR | KS | QbE | IC | SF | | ST | ER | SID | ASV | SD |
| Method | PER↓ | WER↓ | Acc↑ | MTWV↑ | Acc↑ | F1↑ | CER↓ | BLEU↑ | Acc↑ | Acc↑ | EER↓ | DER↓ |
| wav2vec 2.0 (Baevski et al., 2020) | 5.74 | 6.43 | 96.23 | 0.0233 | 92.35 | 88.30 | 24.77 | 14.81 | 63.43 | 75.18 | 6.02 | 6.08 |
| HuBERT (Hsu et al., 2021) | 5.41 | 6.42 | 96.30 | 0.0736 | 98.34 | 88.53 | 25.20 | 15.53 | 64.92 | 81.42 | 5.11 | 5.88 |
| WavLM Base (Chen et al., 2022) | 4.84 | 6.21 | 96.79 | 0.0870 | 98.63 | 89.38 | 22.86 | _20.74_ | 65.94 | 84.51 | _4.69_ | _4.55_ |
| WavLM Base+ (Chen et al., 2022) | 3.92 | _5.59_ | **97.37** | _0.0988_ | 99.00 | **90.58** | 21.20 | **24.25** | **68.65** | **89.42** | **4.07** | **3.50** |
| data2vec (Baevski et al., 2022) | 4.69 | **4.94** | 96.56 | 0.0576 | 97.63 | 88.59 | 25.27 | 17.42 | 66.27 | 70.21 | 5.77 | 6.67 |
| MR-HuBERT (Shi et al., 2024a) | 4.16 | 5.76 | 96.49 | 0.0787 | 98.68 | 88.96 | 23.59 | 16.94 | 65.53 | 76.35 | 5.87 | 5.96 |
| HuBERT it3 50Hz | 4.84 | 7.13 | 96.01 | **0.1016** | 98.37 | 89.66 | 23.96 | 18.00 | 67.45 | 81.72 | 5.77 | 5.06 |
| HuBERT it3 25Hz | 4.40 | 6.87 | 96.59 | 0.0762 | _99.37_ | 87.96 | 25.21 | 19.47 | _68.32_ | _85.42_ | 5.28 | 4.68 |
| HuBERT it3 50Hz 6-Layer | 6.05 | 8.08 | 96.33 | 0.0814 | 98.44 | 88.18 | 24.74 | 15.87 | 67.21 | 80.10 | 5.28 | 5.27 |
| HuBERT it3 50Hz 18-Layer | 4.44 | 6.18 | 96.53 | 0.0922 | 99.13 | 88.95 | 23.02 | 19.45 | 67.86 | 84.39 | 5.04 | 4.82 |
| SpinHuBERT 50Hz | **3.69** | 6.16 | _97.14_ | 0.0903 | 99.24 | _90.06_ | _22.21_ | 19.62 | 68.08 | 83.34 | 5.34 | 4.93 |
| SpinHuBERT 25Hz | _3.83_ | 6.81 | 97.05 | 0.0935 | **99.53** | 87.54 | 25.41 | 19.89 | 67.66 | 82.89 | 4.73 | 4.91 |

## E.2 SPEECH PROCESSING UNIVERSAL PERFORMANCE BENCHMARK

This section evaluates the SSL models in this paper on the Speech Processing Universal Performance Benchmark (SUPERB) (Yang et al., 2021; Tsai et al., 2022). SUPERB is a benchmark that assesses the usefulness of hidden representations of pre-trained speech SSL encoders by applying these representations to a wide range of speech processing tasks. During downstream task training, a speech encoder is frozen, and hidden layer features are extracted. A learnable weighted-sum mechanism then aggregates each frame across all layers to a sequence. The aggregated features are then fed to a lightweight prediction head optimized with a supervised objective like CTC. We encourage the readers to refer to the original SUPERB papers for a more complete explanation of the tasks and evaluation metrics. We follow the implementation in the S3PRL library.[18] The results are shown in Table 21.

Comparing the HuBERT it3 models, we found increasing the number of layers improves almost all tasks (6 vs. 12 vs. 18 layers), showing the effect of model size scaling in downstream tasks. When comparing 50Hz HuBERT it3 and SpinHuBERT models, the 25Hz models are usually better at content-related tasks like PR, ASR, keyword spotting (KS), and speech translation (ST), which might be a result of the shortened sequence of features. The ASR results of HuBERT it3 and Spin-HuBERT are worse than some prior methods, which is consistent with the findings in the previously discussed ASR experiments. Similar to the results in Chang et al. (2023), the SpinHuBERT models trained with Spin units offer high-quality representations and improve content-related tasks like PR. Although trained with speaker-invariant targets, SpinHuBERT performs similarly to prior methods on speaker-related tasks because the speaker information is preserved at the bottom layers, which will be discussed in the next paragraph.

We visualize the weighted-sum mechanism of the SUPERB downstream models in Figure 7 to understand the importance of each hidden layer for different downstream tasks. Following Chang et al. (2022), we normalize the weights by scaling with a factor of the averaged $L^2$ norm of each layer's hidden representations. We observed that top layers are more important for content and semantic-related tasks since the weights have higher values (layers eight to twelve), consistent with prior studies (Chang & Glass, 2024; Yang et al., 2024). Moreover, emotion recognition (ER) relies on the top layers as well (Figure 7g), indicating that classifying speech emotion depends on the content representations. In contrast to the previous tasks, speaker-related problems use the bottom layers, implying that speaker information is stored at those layers (Figures 7h, 7i, and 7j). Comparing Hu-BERT and SpinHuBERT models, the proposed SpinHuBERT models encode speaker information at lower layers (Figures 7h and 7i), which is caused by the speaker-invariant pseudo labels for SSL pre-training, forcing the models to drop speaker information at early layers. Hence, the results verify the findings in the previous paragraph. Overall, this section comprehensively discusses the effectiveness of SpinHuBERT's downstream applications and analyzes the importance of each layer.

---

[18]https://github.com/s3prl/s3prl

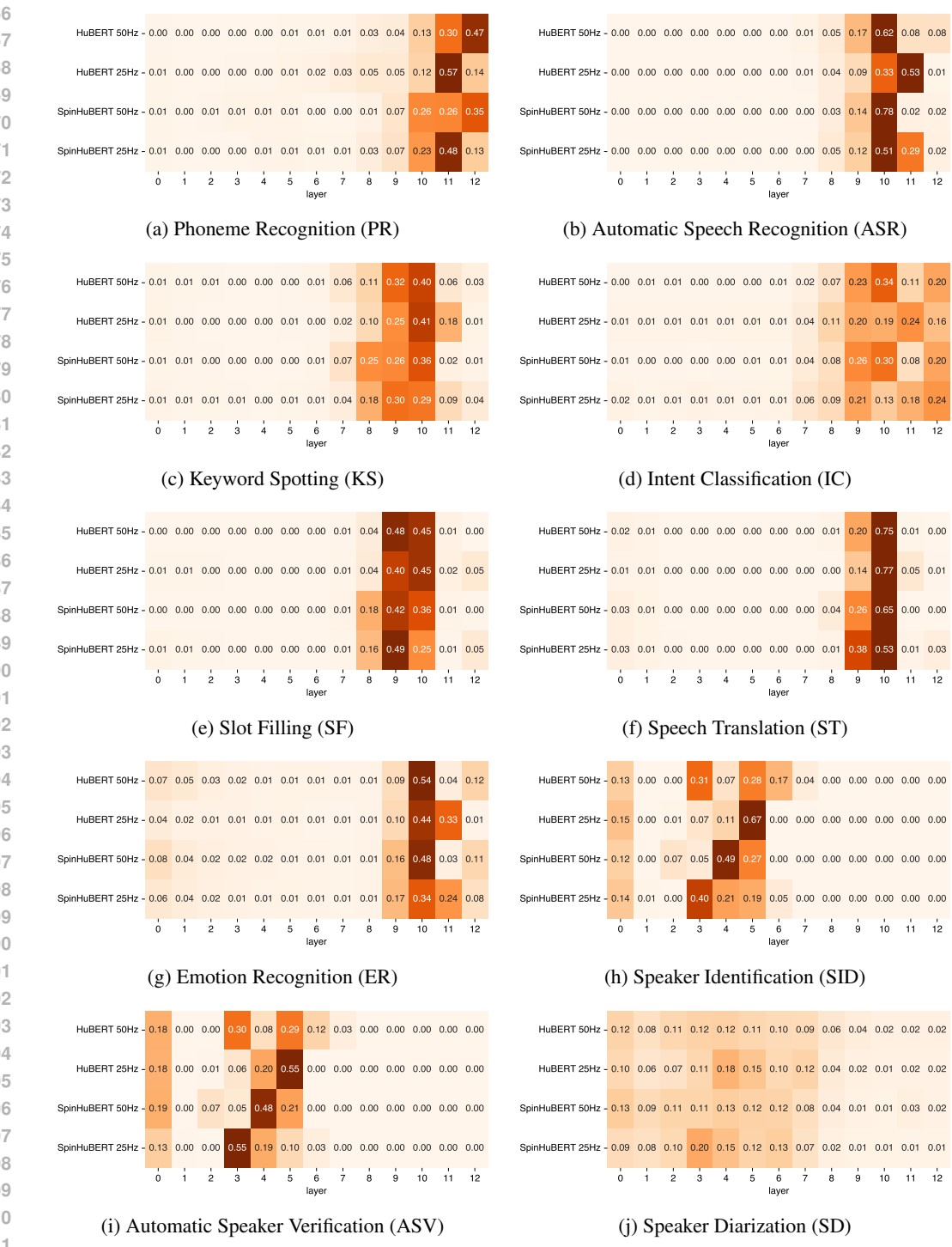

Figure 7: A visualization of the weighted sum mechanism of various SUPERB tasks. The weights are normalized by the averaged $L^2$ norm of each layer's hidden representations. The HuBERT models are the HuBERT it3 Base models pre-trained with 124k hours of speech. The zeroth layer indicates the CNN feature extractor.

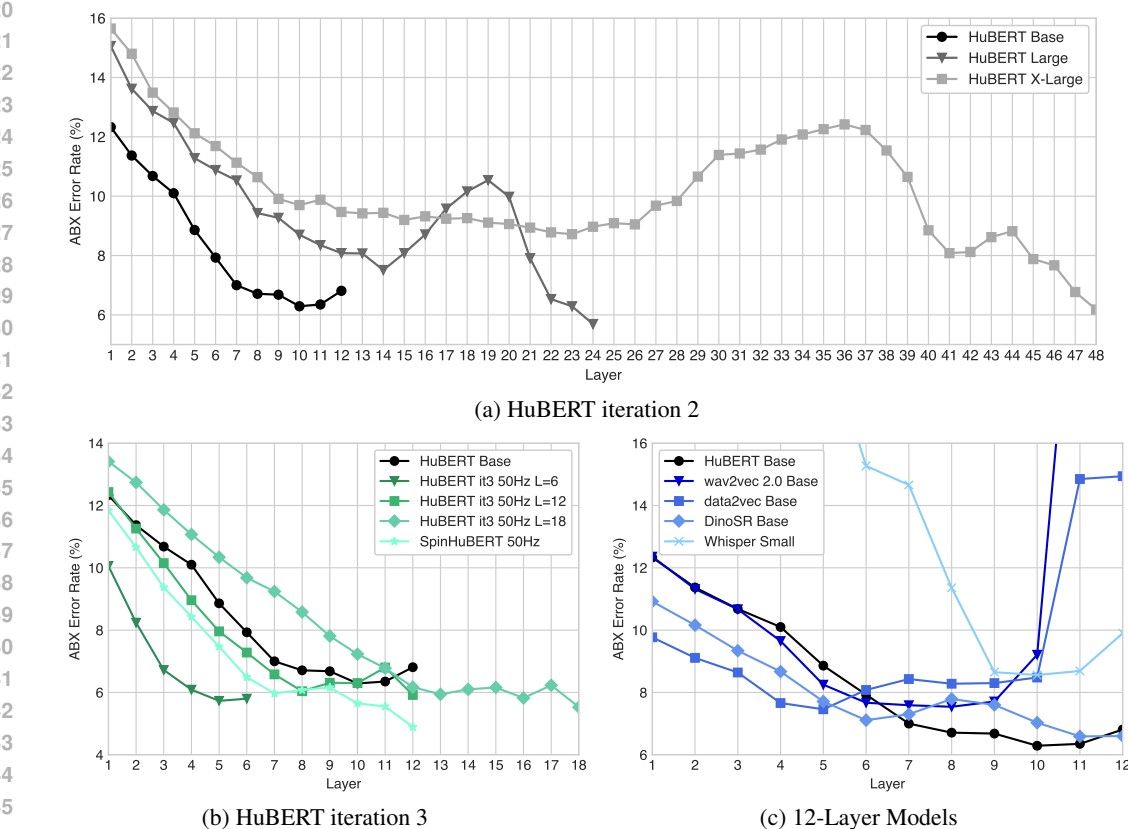

(a) HuBERT iteration 2

(b) HuBERT iteration 3

(c) 12-Layer Models

Figure 8: Layer-wise ABX error rates of SSL speech encoder representations. Each value is an average over the LibriSpeech and Fisher datasets.

### E.3 LAYER-WISE PHONETIC ABX

This section discusses the phonetic representations in several speech SSL models by showing each layer's phonetic ABX error rates. The ABX scores are averaged over dev sets of the LibriSpeech and Fisher datasets (Cieri et al., 2004).[19] Because the Fisher dataset is noisier than LibriSpeech, including this corpus helps simultaneously assess the robustness of these SSL models.

As shown in Figure 8a, the behavior of the Large and X-Large HuBERT models are slightly different than the Base model. The first difference is the lowest ABX layer of the Large and X-Large models, which is at the last, while the Base model is at the 10th because the former two models are trained with the 9th HuBERT Base layer K-means units. Second, the ABX scores of some middle layers in Large and X-Large models are higher than other layers. In Figure 8b, we compare HuBERT models trained with three iterations. SpinHuBERT achieves the lowest ABX error rate at the last layer compared with HuBERT models trained with K-means units. We found the HuBERT it3 models with different sizes share a similar trend in ABX over the hidden layers. Furthermore, we compare several SSL encoders with similar architectures and the number of parameters in Figure 8c. The SSL models all have low ABX scores near the last layer, but wav2vec 2.0 and data2vec have significantly higher values in the last two layers. Moreover, because of the training objective, the Whisper encoder is worse than other SSL models at distinguishing phonemes, corroborating the findings in Appendix C.2. To summarize this section, we found that HuBERT is a relatively superior method for capturing phonetic representations, and SpinHuBERT pushes the limit by improving the pre-training target.

---

[19] https://github.com/facebookresearch/libri-light/tree/main/eval

## F  ADDITIONAL ROBUSTNESS RESULTS

Extending Table 5, this section reports the complete results of robustness experiments in Table 22.

Table 22: Complete unit error distance (UED) robustness results. Unless specified otherwise, all tokenizers are based on HuBERT Base. See Section 4.4 for more information.

| Units | Method | Noise | Time Stretch | Reverb | Pitch Shift |
|---|---|---|---|---|---|
| 50 | K-means$_{50}$ | 29.74 | 39.61 | 28.25 | 44.33 |
| | Gat et al. (2023) | 24.67 | 26.89 | 19.89 | 30.22 |
| | NAST$_{50}$ Messica & Adi (2024) | **9.51** | **17.26** | 9.82 | **16.47** |
| | Spin$_{50}$ | 15.23 | 20.27 | 7.02 | 24.19 |
| | DC-Spin$_{50,4096}$ | 15.09 | 19.83 | 6.32 | 24.87 |
| | + ASR$_{1k}$ | 12.65 | 17.68 | **5.71** | 22.67 |
| | + PR$_{1k}$ | 13.69 | 17.84 | 5.98 | 23.09 |
| 100 | K-means$_{100}$ | 31.38 | 41.97 | 30.42 | 48.68 |
| | Gat et al. (2023) | 25.06 | 29.72 | 21.31 | 32.84 |
| | NAST$_{100}$ Messica & Adi (2024) | **10.82** | **17.45** | 10.35 | **18.74** |
| | Spin$_{100}$ | 17.79 | 23.44 | 7.66 | 28.07 |
| | DC-Spin$_{100,4096}$ | 17.13 | 22.95 | 7.47 | 28.48 |
| | + ASR$_{1k}$ | 14.47 | 19.46 | 7.23 | 24.88 |
| | + PR$_{1k}$ | 15.45 | 19.61 | **7.17** | 25.03 |
| 200 | K-means$_{200}$ | 33.34 | 45.59 | 32.89 | 53.14 |
| | Gat et al. (2023) | 26.76 | 32.99 | 22.94 | 36.45 |
| | NAST$_{200}$ Messica & Adi (2024) | **11.88** | 21.36 | 13.86 | **22.97** |
| | Spin$_{200}$ | 19.95 | 25.12 | 8.97 | 30.70 |
| | DC-Spin$_{200,4096}$ | 19.20 | 24.63 | 8.86 | 30.62 |
| | + ASR$_{1k}$ | 15.97 | **21.05** | **8.13** | 26.45 |
| | + PR$_{1k}$ | 17.37 | 21.88 | 8.86 | 27.39 |
| 500 | K-means$_{500}$ | 36.47 | 50.60 | 39.71 | 58.92 |
| | Gat et al. (2023) | 27.51 | 36.50 | 25.78 | 40.82 |
| | Spin$_{500}$ | 22.33 | 30.52 | 13.80 | 35.87 |
| | DC-Spin$_{500,4096}$ | 21.98 | 29.20 | 13.49 | 35.07 |
| | + ASR$_{1k}$ | 18.92 | 26.12 | 13.89 | 31.48 |
| | + PR$_{1k}$ | 19.73 | 25.95 | 13.37 | 31.30 |
| | SpinHuBERT + DC-Spin$_{500,4096}$ | 18.06 | 24.06 | 11.47 | 25.42 |
| | + ASR$_{1k}$ | 13.47 | **20.34** | 12.25 | **22.35** |
| | + PR$_{1k}$ | 14.23 | 20.53 | **11.29** | 22.74 |
| | + ASR$_{3k}$ | **11.52** | 21.63 | 13.49 | 24.55 |
| | + PR$_{3k}$ | 13.50 | 21.61 | 11.53 | 24.12 |
| 1024 | EnCodec (Défossez et al., 2023) | 82.21 | 84.95 | 87.68 | 97.32 |
| | SpeechTokenizer (Zhang et al., 2024) | 57.09 | 66.29 | 44.12 | 80.66 |
| | Spin$_{1000}$ | **25.38** | **34.15** | **18.98** | **39.45** |

## G  REPURPOSED CHUNK-WISE STREAMING

### G.1  METHOD

This section describes the chunk-wise streaming token extraction approach. As depicted in Figure 9, the tokenizer extracts units after the speaker completed speaking $T_{chunk}$ seconds of speech. Then, after another $T_{shift}$ seconds, the tokenizer repeats the same operation with an increased context length ($T_{chunk} + T_{shift}$ seconds). Assuming the tokenizer produces $L_{chunk}$ and $L_{shift}$ tokens given $T_{chunk}$ and $T_{shift}$ seconds of speech, respectively. Let $L_{overlap} = (L_{chunk} - L_{shift}) / 2$. The last $L_{overlap}$ extracted tokens are neglected in each chunk because the lack of future frames degrades token quality. Under this setup, the tokens extracted in the first few chunks might be less accurate than the offline extracted tokens but will gradually improve after an expanded context.

For instance, in Figure 9, $L_{chunk} = 7$ and $L_{shift} = 3$, making $L_{overlap} = 2$. The first chunk extracts seven tokens and neglects the last two, and the first five tokens are fed into the SLM. When shifted by $T_{shift}$ seconds, the token sequence length increased by three. We then neglect the last two tokens and take the three tokens before them as the SLM input.

We implement the chunk-wise streaming tokenization by repeatedly increasing the audio samples fed into the tokenizer. Each time, the tokenizer (e.g., HuBERT) takes a longer input, and we select tokens according to the methods in the previous paragraph. We then concatenate all selected tokens into one sequence, with approximately the same amount as in the offline extracted sequence. Note that more advanced model architectures and streaming strategies can be implemented to improve streaming performance. For example, reusing encoded hidden states and attention maps from the previous chunks might increase inference efficiency. Nevertheless, we only consider this repurposing approach to achieving streaming tokenization to demonstrate that the proposed tokenizers are streamable without retraining.

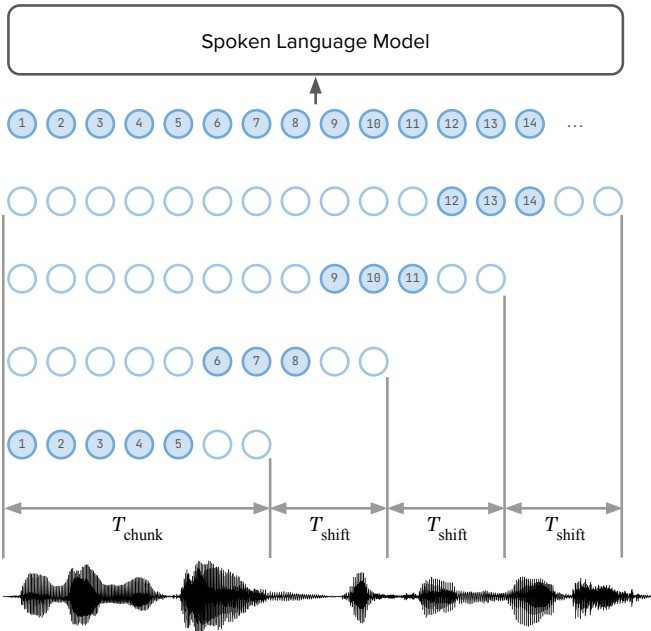

Figure 9: Illustration of chunk-wise streaming token extraction with increasing context length.

**Latency**  The latency calculation is the average latency per chunk. As shown in Table 7, the latency is lower than 20ms, which is about 5% of $T_{shift} = 0.4$s. Moreover, the latency of the last few chunks is about 60ms when a sequence length is 30 seconds long, still significantly lower than the $T_{shift}$, showing that the user experience mainly depends on $T_{chunk}$ and $T_{shift}$, not the tokenizer. Hence, we focus on whether a tokenizer retains the performance when repurposed to streaming mode rather than the latency.

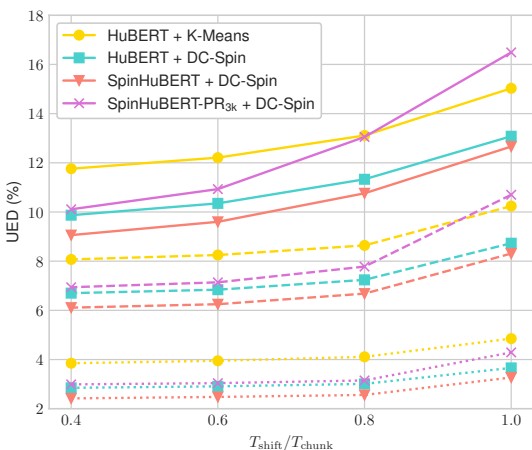

Figure 10: UED between tokenizing offline and chunk-wise streaming. All models are 500-unit tokenizers. Smaller $T_{\text{shift}}/T_{\text{chunk}}$ indicate higher overlap between chunks. Solid, dashed, and dotted lines depict $T_{\text{chunk}} = 1, 2$, and 5 seconds, respectively.

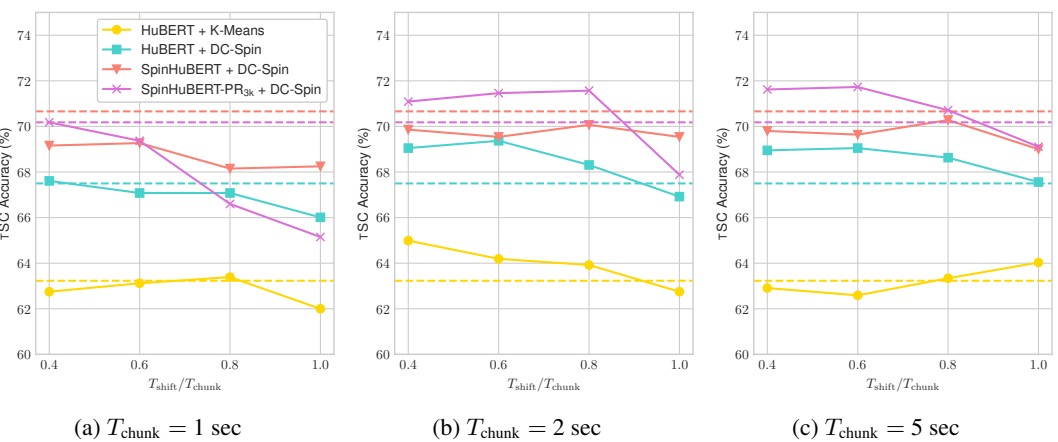

(a) $T_{\text{chunk}} = 1$ sec          (b) $T_{\text{chunk}} = 2$ sec          (c) $T_{\text{chunk}} = 5$ sec

Figure 11: TSC accuracy under different chunk-wise streaming setups. All models are 500-unit tokenizers. Smaller $T_{\text{shift}}/T_{\text{chunk}}$ indicate higher overlap between chunks. Dashed horizontal lines indicate offline tokenizer results.

## G.2 RESULTS

As shown in Figure 10, the UED between offline and streaming tokenization can be significantly reduced by increasing $T_{\text{chunk}}$ (comparing between line styles: solid, dashed, and dotted). The UED can also be improved by reducing $T_{\text{shift}}$ because the higher overlap between chunks ($L_{\text{overlap}}$) prevents using tokens that lack future context. We found that DC-Spin without SFT has a lower UED compared with SFT, which is different from the robustness experiments in Appendix F. A possible cause of this discrepancy is that fine-tuning with PR makes the speech encoder depend on a longer context because the average rate of phonemes is around 10Hz. Still, SpinHuBERT with PR SFT outperforms the HuBERT K-means baseline.

In Figure 11, we show the TSC accuracy under several streaming settings. Similar to the findings in Figure 10, higher $T_{\text{chunk}}$ and lower $T_{\text{shift}}$ results in better accuracy. Unexpectedly, when $T_{\text{chunk}}$ is two or five seconds and $T_{\text{shift}}/T_{\text{chunk}} \leq 0.8$, streaming sometimes outperforms offline tokenization (solid lines are higher than dashed). We suspect the SLM's robustness to token perturbation causes this phenomenon, but we leave this investigation for future studies.

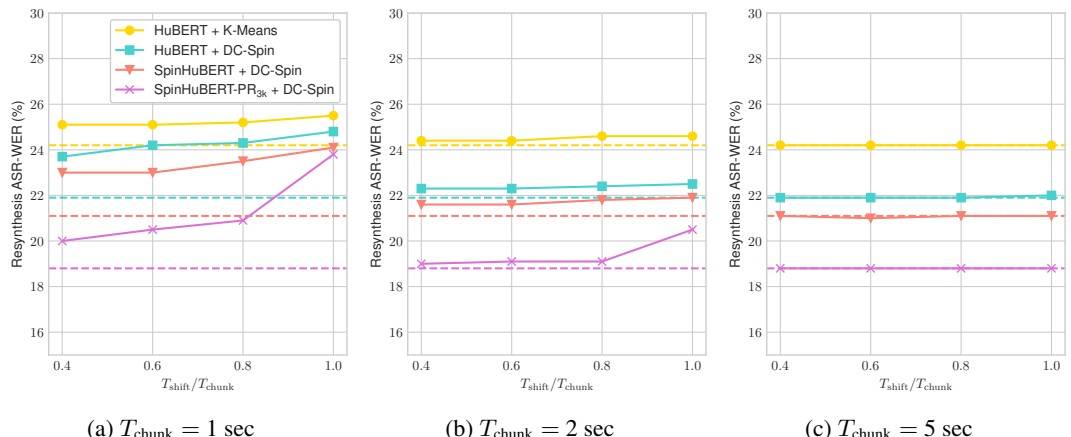

(a) $T_{\text{chunk}} = 1$ sec         (b) $T_{\text{chunk}} = 2$ sec         (c) $T_{\text{chunk}} = 5$ sec

Figure 12: Resynthesis ASR-WER under different chunk-wise streaming setups. All models are 500-unit tokenizers. Smaller $T_{\text{shift}}/T_{\text{chunk}}$ indicate higher overlap between chunks. Dashed horizontal lines indicate offline tokenizer results.

As for resynthesis with streaming tokenization, results in Figure 12 show a similar trend as in UED and TSC experiments. Unlike TSC, streaming tokenization always underperforms offline, indicating that speech resynthesis still requires accurate speech tokens. Overall, experiments in this section demonstrate the possibility of repurposing offline tokenizers to streaming mode with a small performance drop. The findings offer insights into designing speech tokenizers for real-world applications.

# H   CORRELATION BETWEEN METRICS

Figure 13: Pearson correlation coefficients between evaluation metrics computed with tokenizers operate at a 50Hz framerate with 500 units. The upper right corner is the same as Figure 3.

This section reports more results on the correlation between proxy tasks and downstream performance for reference. As shown in the upper left corner of Figure 13, some proxy tasks have high correlations, like ABX and PNMI, but correlate differently with downstream metrics. This observation implies similar proxies should all be considered when predicting a tokenizer's downstream performance. In the lower right corner of Figure 13, most downstream tasks are highly correlated, showing that speech tokenizers usually improve all tasks simultaneously. Furthermore, SLM-based ASR correlates with all other tasks with coefficients higher than 0.74. Therefore, ASR with speech tokens can also serve as a hint to other downstream task performance.

We only compare tokenizers with 500 units operating at a 50Hz framerate because some metrics like ABX error rate are incomparable when the number of units or framerate differ. Perhaps future studies can focus on revealing the relationship between unit size and framerate.

Additionally, Figures 14, 15, 16, and 17 visualize the correlation between these proxies and five major evaluation metrics.

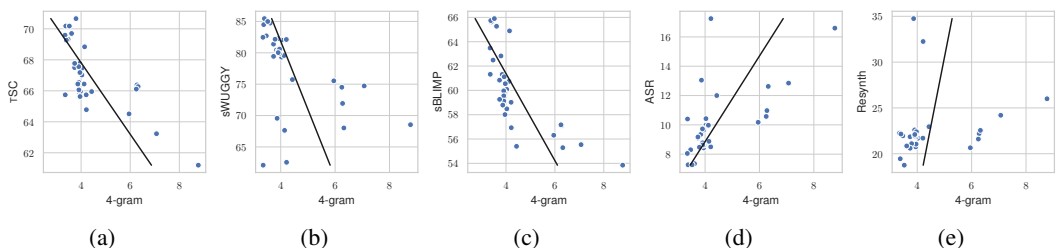

Figure 14: 4-gram predictability (perplexity) vs. zero-shot SLM tasks and SLM-based ASR. Each dot in a plot indicates a 500-unit tokenizer operating at 50Hz.

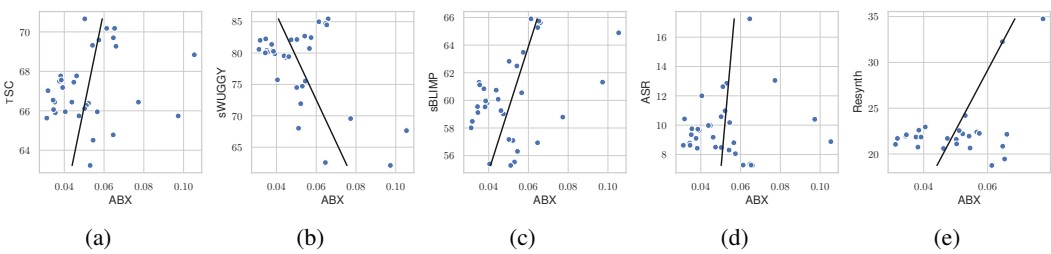

Figure 15: ABX error rate vs. zero-shot SLM tasks and SLM-based ASR. Each dot in a plot indicates a 500-unit tokenizer operating at 50Hz.

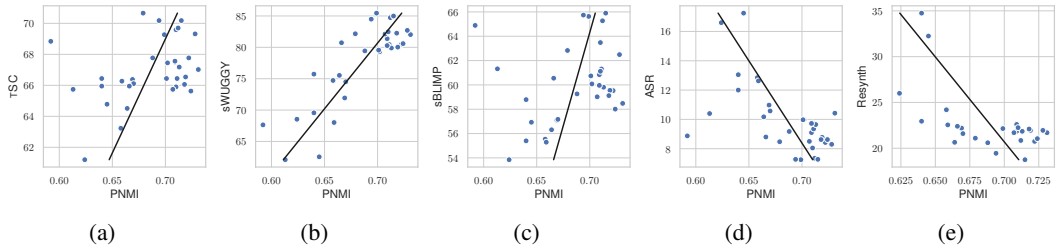

Figure 16: PNMI vs. zero-shot SLM tasks and SLM-based ASR. Each dot in a plot indicates a 500-unit tokenizer operating at 50Hz.

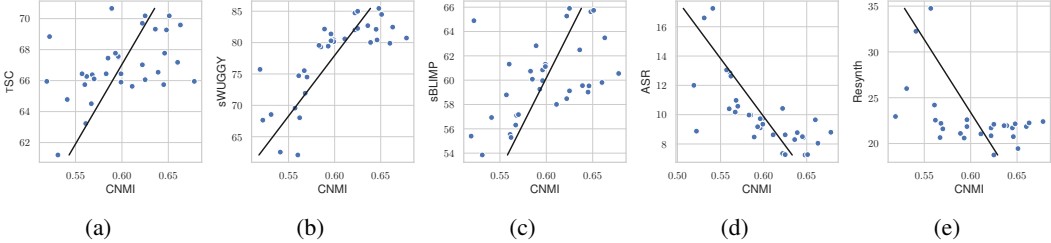

Figure 17: CNMI vs. zero-shot SLM tasks and SLM-based ASR. Each dot in a plot indicates a 500-unit tokenizer operating at 50Hz.

