# OpenReview forum: "DC-Spin: A Speaker-invariant Speech Tokenizer For Spoken Language Models"
_ICLR.cc/2025/Conference — Submitted to ICLR 2025_

### Official Review · Reviewer_wRzi · 2024-10-27

**Soundness:** 3
**Presentation:** 4
**Contribution:** 2
**Rating:** 5
**Confidence:** 4

**Summary:**

This paper proposed a new method for speech tokenization named SpinHuBERT, this tokenization is based on pre-training a HuBert speech encoder with the spin objective, then with an ASR target, and finally fine-tuning with a DC-Spin objective. The evaluation shows the high performance of the method across a wide range of tasks, such as spoken language modeling, speech resynthesis, robustness under distortions and inference efficiency. Furthermore, the authors evaluate different proxy tasks for spoken language modeling which identifies decent candidates.

**Strengths:**

1. The proposed method is novel, and since speech tokanization is widely used improvement are significant.
2. The paper is clearly written which enables to easily re-implement and use the proposed method.
3. The proposed method shows a strong performance over the other baseline methods.

**Weaknesses:**

1. Proposed method out-performs the baseline on most tasks, but the margins are small while the proposed method is significantly more complex.
2. Evaluation on directly synthesizing speech is not performed. It is unclear if in a real setting the proposed method is able to generate coherent and intelligible speech. This evaluation should be added to show the methods performance in the task SpeechLMs are used for.
3. Throughout the paper, UTMOS is used instead of real human raters. While UTMOS is a good proxy, it does not fully replace human raters. At least some of the evaluations should be done using MOS with human raters.
4. Table 3 shows evaluation of other method with much larger models. While a 150m parameter model is decent, SpeechLMs are regularly scaled to larger sizes. The paper is lacking an evaluation of a full-sized SpeechLM model. Such a model would give the reader a better option to appreciate the performance of the proposed method.
5. While the tokens are claimed to be speaker invariant, this is not evaluated in the experimental setup. I would expect the authors to include an experiment which verifies this claim.
6.

**Questions:**

1. In table 8 and 4 ASR-WER is very high. What ASR model was used? What is the reason for this?
2.  Why is using the proxy tasks better than directly using TSC/ sWUGGY / sBLIMP. They don't seem easier to compute and TSC/ sWUGGY / sBLIMP are already proxies for the ability to generate intelligible audio.
3. Why use HuBert as the base encoder and not some other encoder? what are the properties of HuBert that make it best for this task?
4. Why fine-tune and not train from scratch with the SPIN objective?
5. In figure 1, we see that different tasks have the best performance with different code-book sizes. For example the 1000 size codebook has the best performance on TSC (but is not used later in the paper). How should the final model be picked?

---

> ### Author Response · Authors · 2024-11-14
> **Response to Reviewer wRzi**
>
> We sincerely appreciate the reviewer's detailed feedback and the opportunity to address the concerns raised.
>
> **Response to Weakness 1**
> In Table 2, the performance gains of our method are notable relative to prior work. The speech tokenizers proposed in NAST and Gat et al. are more complex, often requiring extensive hyperparameter tuning, while we found Spin and DC-Spin to be more stable, with minimal tuning required.
>
> In Table 3, our approach demonstrates comparable or better results than high-resource speech LMs, even with significantly less data and compute, suggesting that our tokenizer effectively lowers the training requirements.
>
> **Response to Weakness 2**
> We acknowledge the limitation of not including SLM-based TTS evaluations. While time constraints prevent us from conducting this experiment, we reference another paper on self-supervised speech encoder-based tokenizers [1], where consistent results between speech resynthesis and TTS suggest that our speech tokenizer would likely show similar performance in TTS.
>
> **Response to Weakness 3**
> While we understand UTMOS is only a proxy for MOS, policies at some authors' institutions make human MOS evaluations difficult within the rebuttal period.
>
> **Response to Weakness 4**
> We conducted some initial scaling experiments, shown in the table below, which indicate slight improvements with larger models. (tokenizer: HuBERT Base + DC-Spin$_{200,4096}$)
>
> | SLM Size | tSC   | sWUGGY | sBLIMP |
> | -------- | ----- | ------ | ------ |
> | 150M     | 69.21 | 80.59  | 62.13  |
> | 300M     | 69.32 | 81.91  | 62.33  |
>
> However, larger models tend to overfit, particularly given our smaller dataset compared to prior works, which typically use 10X more data. We would like to conduct further scaling experiments, though these may not be feasible within the rebuttal period.
>
>
> **Response to Weakness 5**
> Our DC-Spin training objective aligns with Spin, whose speaker invariance has been demonstrated in [2,3]. We rely on this property in our approach to achieving a speaker-invariant tokenizer.
>
>
> **Response to Question 1**
> Following Expresso [4], we use an ASR based on wav2vec 2.0 Large pre-trained on Libri-Light 60k hours dataset and fine-tuned with LibriSpeech 960 hours corpus (Section 4.3). The high ASR-WER is due to out-of-domain speaking styles in the Expresso dataset, such as laughter and whispering.
>
> **Response to Question 2**
> The proxy tasks in Section 4.7 (bitrate, n-gram predictability, ABX, PNMI, and CNMI) do not need additional SLM training, mitigating the need for 12 hours of training on 8 GPUs. These proxies can be computed within an hour, allowing for faster development and evaluation.
>
> **Response to Question 3**
> As explained in Section 3.3 and Table 1, HuBERT offers a relatively better initialization for DC-Spin and has stable training compared with EMA-based data2vec. We suspect that the HuBERT training objective is more similar to Spin and DC-Spin, contributing to the success of these fine-tuning methods, i.e., predicting discrete tokens. The only difference between the Spin objective and HuBERT is that Spin's prediction targets are learned during training, while HuBERT has fixed targets.
>
> **Response to Question 4**
> In [3], the authors have shown that fine-tuning more layers leads to model collapse. We also observed a similar pattern in our experiments. This phenomenon is caused by a trivial solution to the Spin objective that can be easily achieved. The trivial solution is to ignore all input speech information and possibly encode only the relative position of each frame. Hence, freezing some layers of a pre-trained model enforces the encoder to preserve the original information of speech.
>
> **Response to Question 5**
> The nature of each downstream task has led us to choose the final DC-Spin codebook size carefully. Here are some properties in the tokenizer required for each task.
> 1. **tSC** and **sBLIMP** benefit from smaller codebooks for coarse-grained semantic and syntactic information.
> 2. **sWUGGY** and **Resynthesis** require finer phonetic and acoustic details, favoring a larger codebook.
>
> According to Figure 1, we chose 500 to obtain the best overall performance across tasks. Note that this choice is based on the downstream tasks and the tokenizer's operating framerate (50Hz), a different setup requires a different codebook size.
>
> ---
>
> [1] Shi, Jiatong, et al. "MMM: Multi-Layer Multi-Residual Multi-Stream Discrete Speech Representation from Self-supervised Learning Model." arXiv (2024).
> [2] Chang, Heng-Jui, Alexander H. Liu, and James Glass. "Self-supervised fine-tuning for improved content representations by speaker-invariant clustering." arXiv (2023).
> [3] Chang, Heng-Jui, and James Glass. "R-Spin: Efficient Speaker and Noise-invariant Representation Learning with Acoustic Pieces." arXiv (2023).
> [4] Nguyen, Tu Anh, et al. "Expresso: A benchmark and analysis of discrete expressive speech resynthesis." arXiv (2023).

---

> > ### Comment · Reviewer_wRzi · 2024-11-26
> >
> > Thanks you for providing clarifications.
> > First, some of the answers to my questions are satisfying (such as q1, q3, q4, q5) these should be clearly explained in the paper.
> > Second, regarding W3, W2 and Q1, the authors response is not satisfying. I understand that running MOS experiments during the rebuttal is not trivial, but these are a standard measure in speech and should have been available in the original paper. This, combined with the low WER, expressive dataset and reliance on previous work for the evaluation of speech quality are not satisfying.  In the case of the expressive dataset, the unreliable WER increases the importance of a human evaluation of speech quality.

---

### Official Review · Reviewer_SENG · 2024-10-30

**Soundness:** 1
**Presentation:** 2
**Contribution:** 1
**Rating:** 1
**Confidence:** 4

**Summary:**

This paper introduces Double-Codebook Speaker-invariant Clustering (DC-Spin) as a method to enhance speech tokenization by making the speech tokens to be speaker-invariant.
By decreasing the proportion of the speaker information in the speech tokens, DC-Spin aims to improve zero-shot SLM tasks and speech resynthesis. Through comparisons of tokenization techniques and model scalability, the study highlights the effectiveness of tokens easily modeled by n-gram language models or aligned with phonemes, offering valuable guidelines for designing speech tokenizers for SLMs.

**Strengths:**

This paper proposes a method called DC-Spin to improve speech tokenization.
The speech tokens generated by the proposed method is speaker-invariant and can preserve more content information compared to open-source tokenizers.

**Weaknesses:**

The motivation of this paper does not sound convincing.
A common way of connecting a speech encoder to a LM is directly through speech representation, which is of continuous values.
The discretization from representation to discrete tokens will only degrade the quality of the speech input. Check out the papers below:
https://arxiv.org/pdf/2402.08846
https://arxiv.org/abs/2309.00169
https://www.isca-archive.org/interspeech_2024/yang24f_interspeech.pdf
The discretized tokens will eventually be converted to continuous vectors through the embedding table of the LM.
Therefore, there is no need to discretized the continuous embedding for the input of LM.

From the LM output perspective, yes, there is a need for speech tokenization if we want the LM to generate speech.
However, the TTS community has already gone far beyond word accuracy. We are talking about voice clone, high naturalness.
Thus, removing the speaker information may even hinder these speech tokens to be used.
Even if we neglect the voice clone issue, the ASR-WER of the speech resynthesis is far below standard (18.x in the paper). That means the system can't be used. Checkout the numbers in the following papers:
https://arxiv.org/abs/2309.00169
https://arxiv.org/abs/2406.02430
https://arxiv.org/abs/2301.02111

**Questions:**

The Spin mentioned in the paper perturb the speech utterance by randomly scaling the F0 and formant frequencies.
As I know, speaker variation is more than different F0 and formants. How do you handle speaking style, which is also a kind of speaker-variant?

In the most right hand side of Figure 2, it is not clear to me how the codebook is formed and how the discrete speech tokens are generated? Is K-means clustering still be used?

---

> ### Author Response · Authors · 2024-11-14
> **Response to Reviewer SENG**
>
> We sincerely appreciate the reviewer's detailed feedback and the opportunity to address the concerns raised. We believe, however, that certain assumptions may not fully capture the recent advances and diversity in spoken language model (SLM) research, as a range of approaches for integrating speech into LMs are actively being explored.
>
> ---
>
> **(I) Response to the Reviewer's Comments on Speech Tokenization**
> The reviewer suggests that connecting a speech encoder to an LM should involve continuous speech representation, not discrete tokens. While continuous representations are indeed common, recent work has shown that discrete token-based methods are both valuable and effective. For example:
> 1. *Spirit LM* [1]: Uses K-means clustered units in training a speech and text interleaved LLM.
> 2. *Moshi* [2]: Proposes Mimi, a neural audio codec that transforms speech into discrete tokens for their full-duplex spoken dialogue framework's input and output units.
> 3. *hertz-dev* [3]: Similar to Moshi, also uses a neural audio codec.
>
> Moreover, a recent study comparing discrete and continuous representations for SLM-based ASR found both approaches to offer comparable performance [4]. Additionally, one of the reviewer's own references (https://arxiv.org/abs/2309.00169) supports the use of discrete tokens in ASR, reinforcing our approach. We hope this clarifies the value of discrete tokens in SLM.
>
> ---
>
> **(II) Response to the Reviewer's Comments on Speech Generation**
> The reviewer notes that modern TTS focuses on high naturalness and voice cloning, expressing concern that removing speaker information could reduce effectiveness. However, the referenced papers also use discrete tokens as TTS output units, which seems to contradict the concern raised. Furthermore, in Tables 4 and 19 of our paper, we show that our speech tokens outperform neural audio codecs under similar bitrates.
>
> The ASR-WER in our work is higher due to two main factors:
> 1. We used expressive speech data, such as laughing and whispering, which is not standard in ASR training and naturally yields higher WER.
> 2. The vocoder we used was not carefully optimized, as our focus was on demonstrating the tokenizer's effectiveness. Future research could improve resynthesis quality using enhanced vocoder architectures (e.g., a diffusion transformer).
>
> Additionally, regarding voice cloning, the goal of removing speaker information is to enhance content representation; speaker details can be reintroduced in the vocoder stage, as outlined in Section 4.1.
>
> ---
>
> **Response to Question 1**
> > How do you handle speaking style, which is also a kind of speaker-variant?
>
> Our approach removes speaker identity from speech tokens to encourage the tokenizer to encode better content information. We did not consider perturbing speaking styles like loudness and emotional speech. Hence, we leave this part for future studies because these kinds of perturbations require a more complex data augmentation / generation process. E.g., controllable voice conversion or TTS.
>
>
> **Response to Question 2**
> > In the most right hand side of Figure 2, it is not clear to me how the codebook is formed and how the discrete speech tokens are generated? Is K-means clustering still be used?
>
> As described in Section 3.2, we first explained that we directly use the tokens from a Spin codebook without K-means. Next, we described that DC-Spin has two Spin codebooks and share the same encoder. We can add more detailed illustration in the appendix to clarify how DC-Spin works.
>
> ---
>
> **We believe the weaknesses mentioned do not constitute grounds for rejecting the paper. While we respect the practical perspective of the reviewer, we encourage the reviewer and Area Chair to consider the academic contribution and future impact of our work.**
>
> ---
>
> [1] Nguyen, Tu Anh, et al. "Spirit-lm: Interleaved spoken and written language model." arXiv preprint arXiv:2402.05755 (2024).
> [2] Défossez, Alexandre, et al. "Moshi: a speech-text foundation model for real-time dialogue." arXiv preprint arXiv:2410.00037 (2024).
> [3] https://si.inc/hertz-dev/
> [4] Xu, Yaoxun, et al. "Comparing Discrete and Continuous Space LLMs for Speech Recognition." arXiv preprint arXiv:2409.00800 (2024).
> [5] Shi, Jiatong, et al. "MMM: Multi-Layer Multi-Residual Multi-Stream Discrete Speech Representation from Self-supervised Learning Model." arXiv preprint arXiv:2406.09869 (2024).

---

> > ### Comment · Reviewer_SENG · 2024-11-19
> >
> > While the authors list several recent works that utilize discrete tokens, they fail to provide a concrete reason why discrete tokens should be used instead of representations in the ASR perspective, especially when these discrete tokens are derived from representations.
> > I referenced the RepCodec paper (https://arxiv.org/abs/2309.00169) because it presents experiments demonstrating that using discrete tokens performs worse than using representations in ASR tasks. However, the RepCodec paper itself does not offer a clear rationale for favoring discrete units over representations.
> > Merely suggesting references does not imply agreement with the work's motivation.
> > Why should one go through the process of converting representations into discrete tokens and then back into representations? If this question remains unanswered, it calls into question the value of using speech discrete tokens in ASR.
> > If this fundamental question cannot be addressed, it leads me to perceive that speech discrete tokens may hold little value in the realm of ASR.
> >
> >
> >
> > I would like to emphasize that I agree with the importance of using discrete tokens in the TTS perspective. It is a crucial step for the language model to generate speech/audio. I recommend the authors to consider the ASR-WER numbers in the cited papers: below 7 in RepCodec (https://arxiv.org/abs/2309.00169), below 6 in Vall-E (https://arxiv.org/abs/2301.02111), and below 3 in Seed-TTS (https://arxiv.org/abs/2406.02430).
> > All of these studies demonstrate the ability to achieve low ASR-WER and high speaker similarity simultaneously, setting a standard for the community. Hence, I find no contradiction between my references and the raised concerns.
> > If the aim of removing speaker information is to enhance content representation, why do the authors utilize speech data containing a significant amount of laughter and whispering? I believe that the speech content is not well preserved with such a high ASR-WER (18.x). The authors should showcase their method within a more reasonable range of ASR-WER.

---

> > > ### Author Response · Authors · 2024-11-23
> > >
> > > The units learned by Spin are designed to be more **content-based**, with a strong relationship to underlying **phonemic patterns**. Consequently, the notion of discrete tokens is reasonable, as it aligns with the finite nature of phoneme inventories in any language. While the usefulness of discrete tokens for generation tasks, such as TTS, is well-recognized, we argue that the same representations can also hold value for understanding tasks like ASR. These two processes share a **common underlying linguistic representation**, and exploring shared representations for both tasks is, in our view, a valid and promising direction.
> > >
> > > Furthermore, we believe it is premature to conclude that discrete representations lack utility for ASR and to dismiss research investigating this approach. Discretization can be understood as **information compression**, which has demonstrated success in a wide range of deep learning models for speech. While the RepCodec paper highlights certain limitations, it does not provide a conclusive rationale against the use of discrete tokens. Instead, it underscores the need for further exploration, which is precisely what our work aims to address.
> > >
> > > We appreciate your acknowledgment of the importance of discrete tokens for generation tasks and hope that our clarification sheds light on their potential merits for understanding tasks as well. Thank you for engaging with our work and for contributing to this important discussion.

---

### Official Review · Reviewer_eraa · 2024-11-03

**Soundness:** 3
**Presentation:** 4
**Contribution:** 3
**Rating:** 8
**Confidence:** 2

**Summary:**

This paper presents the development of robust and efficient speech tokenizers for spoken language modeling and speech resynthesis. The authors introduce two innovative methods, SpinHuBERT and DC-Spin, designed to meet four essential criteria for ideal tokenizers: capturing phonetic information, preserving acoustic details, robustness to perturbations, and enabling efficient inference. Experimental results demonstrate that a 150M parameter Transformer decoder, trained on downstream tasks, performs optimally with DC-Spin speech tokens. Furthermore, DC-Spin delivers superior reconstruction quality at extremely low bitrates compared to popular codecs. The authors also identify proxy metrics — shown to correlate highly with downstream task performance — and advocate for these metrics as a basis for evaluating future speech tokenizers.

**Strengths:**

* **Originality:** The paper’s contribution of SpinHuBERT and DC-Spin tokenizers demonstrates originality.
* **Quality:** The evaluation is comprehensive, evaluating the proposed tokenizers across multiple downstream tasks and metrics to show their effectiveness.
* **Clarity:** The presentation is clear.
* **Significance:**  Disentangling factors within discrete speech tokens, as explored in DC-Spin, addresses a growing area of interest (e.g., FACodec). DC-Spin seems to stand out as one of the first speech tokenizers to separate speaker information from codebooks.

**Weaknesses:**

**Comparison in Speech Resynthesis:** While the results show that DC-Spin performs well at low bitrates (<1.5 kbps), it is worth noting that current state-of-the-art speech synthesis systems (e.g., NaturalSpeech2, VoiceCraft, MaskGCT) use codecs with higher bitrates to achieve high-quality speech reconstruction. A comparison of DC-Spin against existing tokenizers at these higher bitrates would provide valuable insight into its performance for speech resynthesis under more typical conditions.

**Questions:**

1. **Claim:** “The first codebook (primary) extracts discrete units for downstream applications. The second codebook (auxiliary) is a large codebook that enhances the encoder’s capability to capture fine-grained phonetic units.” Might there be an ablation study on the independent use of each codebook to support this finding? (i) Evaluating downstream task performance using only the primary codebook vs. only the auxiliary codebook, (ii) analyzing the phonetic content captured by each codebook independently, and (iii) examining how varying the size of each codebook impacts the overall system performance are useful experiments to help validate the claim on the role of codebooks.
2. **Robustness Testing:** Could the authors provide further details on the types and degrees of perturbations applied in testing the robustness of DC-Spin? Specifically, can you provide quantitative details on the noise levels, time stretch factors, reverberation parameters, and pitch shift ranges, and comment on how well they compare to real-world conditions?
3. Can DC-Spin can be adapted to operate at higher bitrates comparable to systems like NaturalSpeech2 or VoiceCraft? If so, how does its performance compares in terms of speech quality metrics (e.g. MUSHRA scores) at those higher bitrates?

---

> ### Author Response · Authors · 2024-11-14
> **Response to Reviewer eraa**
>
> We sincerely appreciate the reviewer's thorough feedback and are grateful for the opportunity to clarify and address these points.
>
>
> **Response to Question 1**
> We observed the auxiliary codebook in DC-Spin behaves similarly to the single large codebook used in the original Spin model, with comparable quality in SLM and codebook metrics. However, the primary codebook's performance has improved in DC-Spin (Table 14). Additionally, we have conducted an ablation study with varying auxiliary codebook sizes, as detailed in the revised paper's Appendix B.1 (Figure 4). These results underscore the auxiliary codebook's effectiveness.
>
>
> **Response to Question 2**
> We follow the robustness evaluation pipeline in [1] (code at https://github.com/ShovalMessica/NAST), and we followed their official setup as described in Section 3 of [1]:
> > (i) time-stretch using a Phase Vocoder method to stretch or shrink the time domain signal in the range $[0.8, 1.2]$ without changing the pitch; (ii) pitch-shifting the speech signal by four semitones using the resampling method over the time-stretched signal; (iii) reverberation following similar setting as simulated via the pyroomacoustics audio room simulations package; and (iv) noise injection using a randomly sampled Signal-to-Noise Ratio (SNR) in the range of $[5, 15]$. Background noises are sampled from the Deep Noise Suppression (DNS) challenge which includes a diverse set of noise types from AudioSet, Freesound, and Demand.
>
>
> **Response to Question 3 and the Weakness**
> To explore DC-Spin's potential at higher bitrates, we experimented with residual vector quantization (RVQ) to expand bandwidth and incorporated Mel spectrogram reconstruction training (Appendix B.3). We applied the Spin loss to the primary codebook to enhance speaker-invariant content, while additional codebooks were designated for capturing fine-grained acoustic details. Despite these modifications, zero-shot SLM performance did not improve significantly, though we believe further studies could yield higher resynthesis quality. A possible direction for future work is outlined in [2].
>
> ---
>
> [1] Messica, Shoval, and Yossi Adi. "NAST: Noise Aware Speech Tokenization for Speech Language Models." arXiv preprint arXiv:2406.11037 (2024).
> [2] Shi, Jiatong, et al. "MMM: Multi-Layer Multi-Residual Multi-Stream Discrete Speech Representation from Self-supervised Learning Model." arXiv preprint arXiv:2406.09869 (2024).

---

> > ### Comment · Reviewer_eraa · 2024-11-26
> >
> > Thank you for the detailed responses. I am satisfied with the clarifications and results. I would like to see these details included in the paper to enhance its clarity and impact. I find the work novel and maintain my original score of 8. I look forward to seeing how this work evolves.

---

### Official Review · Reviewer_q3wD · 2024-11-04

**Soundness:** 3
**Presentation:** 2
**Contribution:** 2
**Rating:** 5
**Confidence:** 5

**Summary:**

The paper presents a novel approach to speech tokenization for spoken language models (SLMs), focusing on speaker invariance and phonetic richness through the Double-Codebook Speaker-invariant Clustering (DC-Spin) technique. DC-Spin utilizes a dual codebook design to separate phonetic and speaker-invariant features, enhancing SLM performance in tasks like automatic speech recognition (ASR) and speech synthesis. Additionally, the paper proposes SpinHuBERT, a pre-trained model that leverages DC-Spin for robust and accurate tokenization. The contributions of DC-Spin are evaluated against existing tokenizers, demonstrating improved robustness, phonetic alignment, and real-time processing efficiency, setting a new standard in resource-constrained and scalable speech tokenization.

**Strengths:**

1. Introduced the Double-Codebook Spin to capture the fine-grained phonetic units better, and gave the detail process about how to select the codebook size.
2. The experimental setup is thorough, covering both zero-shot and supervised tasks, with a clear evaluation on robustness and inference efficiency.
3. This work analyzed multiple proxy tasks on speech tokenizers to reveal their relationship with the performance of the spoken language model.

**Weaknesses:**

1. Although this work states three contributions, they were incremental somehow. For example, the DC-Spin doesn't show convincing advantages against Spin method in Table 2, especially considering the much larger codebook size of DC-Spin. As for the chunk-wise streaming simulation, it has been used in many similar works such as ASR and TTS. And the authors claimed it in the contribution part but they just mentioned it in a short paragraph in the experiment section.
2. The compared neural audio codec methods were a little bit dated. Encodec and SpeechTokenizer methods were proposed in 2022 and 2023. Many new and powerful works have been proposed since then. For example, descript audio codec (DAC) and hifi-codec should be included in the comparison. Besides, WavTokenizer is another work focusing on low bitrate, making it suitable for comparison.
3. The motivation of designing the DC-Spin is not clear. The authors claimed the auxiliary codebook can capture fine-grained phonetic units but didn't provide explanations. Moreover, it both primary and auxiliary codebooks pay attention to the phonetic units, the acoustic detail would be ignored in DC-Spin to some extent, which is harmful for the generative task.

**Questions:**

1. I am a little bit confused about the bitrate of Kmeans and DC-Spin reported in Table 4 and Table 19. For example, DC-Spin_{500,4096} has the primary codebook with 500 size and the auxiliary codebook with 4096 size. But the final bitrate is the same as the Kmeans_{500} which only has a 500 size codebook. Could you explain it in detail?

---

> ### Author Response · Authors · 2024-11-14
> **Response to Reviewer q3wD**
>
> We sincerely appreciate the reviewer's detailed feedback and valuable insights, which have helped us clarify key aspects of our work.
>
>
> **Response to Weakness 1**
> In Table 2, the improvements of DC-Spin over Spin become more pronounced with smaller primary codebook sizes (e.g., units = 50 or 100). While the improvements are less evident at 500 units, they are still present. Additionally, the chunk-wise streaming approach illustrates the ease with which our tokenizer adapts to streaming applications. For this reason, we provided the detailed explanations in the appendix. We will revise the contribution section in the introduction to emphasize this more clearly.
>
>
> **Response to Weakness 2**
> Although the baseline methods we used, EnCodec and SpeechTokenizer, are relatively recent (2022 and 2023) and remain widely used, we acknowledge that there are newer alternatives. We can possibly try to experiment with DAC, but due to the rebuttal time frame, we might not be able to report the results. Additionally, WavTokenizer is considered concurrent work since it was first posted to ArXiv in late August this year (https://iclr.cc/Conferences/2025/FAQ).
>
>
> **Response to Weakness 3**
> As noted in Section 3.2, larger Spin codebooks have been shown by Chang et al. (2023) to better capture phonetic representations. Since both the primary and auxiliary codebooks share the same encoder, we expect the auxiliary codebook to indirectly assist the primary codebook in encoding high-quality phonetic units. We have clarified this mechanism in the paper. For example: "Moreover, Chang et al. (2023) found larger Spin codebooks capture better phonetic representations *in the encoder*." Although we found that DC-Spin helps capturing phonetic units, there is no evidence about ignoring acoustic details. Besides, in Tables 4 and 19, we show that DC-Spin consistently outperform K-means in speech resynthesis.
>
>
> **Response to Question 1**
> The auxiliary codebook is used only during training and is not involved in token extraction. When tokenizing speech with DC-Spin, we rely solely on the primary codebook, resulting in a bitrate comparable to that of K-means$_{500}$.

---

> > ### Comment · Reviewer_q3wD · 2024-11-20
> >
> > Thanks for the explanation of my concerns. However, there are some issues in the current version of this paper.
> >
> > **Suggestions to the paper's structure**
> >
> > The structure of this paper is too messy to capture the core point for me, especially about the proposed method's contributions and pipeline. The biggest problem is that the description of the proposed method in Section 3 is not consistent with the content in the introduction part. Besides, the order of the sub-section in Section 3 is also confusing. How about introducing the SpinHuBert in Section 3.2 and then introducing the proposed DC-Spin in Section 3.3? It is not a good presentation to describe the components in reverse order to the actual components working in the pipeline.
> >
> > **Interpretation and experimentation are contradictory**
> >
> > Although the author explained the motivation of the auxiliary codebook, it is not very convincing. The authors claimed that DC-Spin shows an evident advantage against Spin when the primary codebook size is small (50 or 100) because the auxiliary codebook with a large size in DC-Spin can help the shared encoder extract more phonetic information. However, in the experiment presented in the paper, the size of the primary codebook is 500, which means the improvement compared with Spin is limited because 500 is enough for Spin to extract necessary phonetic information and the auxiliary codebook becomes dispensable.
> >
> > ---
> >
> > Overall, I intend to keep the score unchanged.

---

> > > ### Author Response · Authors · 2024-11-21
> > > **Further Response to Reviewer q3wD**
> > >
> > > We sincerely thank the reviewer for their thoughtful feedback and for taking the time to clarify their concerns.
> > >
> > > **Suggestions to the paper's structure**
> > > Our intention in presenting DC-Spin before SpinHuBERT was to emphasize DC-Spin as the primary contribution, with SpinHuBERT serving as a complementary enhancement. However, we understand the reviewer's concerns regarding the flow and clarity of the section. We will gladly revise the structure as suggested, introducing SpinHuBERT first, followed by DC-Spin, to align better with the pipeline and improve the overall readability.
> > >
> > > **Interpretation and experimentation are contradictory**
> > > We acknowledge the reviewer's concern and wish to clarify further. While our experiments demonstrated that DC-Spin offers substantial improvements with smaller primary codebook sizes (50 or 100), we also observed consistent gains with larger codebooks, including size 500. These results confirm that the auxiliary codebook provides additional benefits regardless of the primary codebook size, although the extent of improvement naturally varies.
> > >
> > > ---
> > >
> > > We deeply appreciate the reviewer's constructive suggestions and remain committed to addressing these points in the final revision.

---

### Meta-Review · Area_Chair_22QL · 2024-12-20

**Metareview:**

The authors propose a method for speech tokenization that is speaker invariant. The paper received four reviews, with three generally negative and one positive. While the paper did receive one vote for acceptance, that vote had low confidence, while the three votes for rejection were much more confident in their evaluation. The reviewers were not too motivated to engage with the paper generally, but one reviewer did note that concerns remained after the rebuttal. In general, it was felt that the evaluation of the propose method should be improved before acceptance to a top-tier conference.

**Additional Comments On Reviewer Discussion:**

There was not much discussion, but one negative reviewer noted that their concerns were not able to be adequately addressed in this review cycle. While one review was a very negative "1", it was more confident and in line with other reviews, while the one overly positive review "8" was much less confident in their evaluation.

---

### Decision · Program_Chairs · 2025-01-22

Reject